# When abstract becomes concrete, naturalistic encoding of concepts in the brain

**Viktor Nikolaus Kewenig\*, Gabriella Vigliocco, Jeremy I Skipper**

Experimental Psychology, University College London, London, United Kingdom

## eLife assessment

Kewenig et al. present a timely and **valuable** study that extends prior research investigating the neural basis of abstract and concrete concepts by examining how these concepts are processed in a naturalistic stimulus: during movie watching. The authors provide **convincing** evidence that the varying strength of the relationship between a word and a particular visual scene is associated with a change in the similarity between the brain regions active for concrete and abstract words. This work makes a contribution that will be of general interest within any field that faces the inherent challenge of quantifying context in a multimodal stimulus.

**\*For correspondence:**
ucjuvnk@ucl.ac.uk

**Competing interest:** The authors declare that no competing interests exist.

**Abstract** Language is acquired and processed in complex and dynamic naturalistic contexts, involving the simultaneous processing of connected speech, faces, bodies, objects, etc. How words and their associated concepts are encoded in the brain during real-world processing is still unknown. Here, the representational structure of concrete and abstract concepts was investigated during movie watching to address the extent to which brain responses dynamically change depending on visual context. First, across contexts, concrete and abstract concepts are shown to encode different experience-based information in separable sets of brain regions. However, these differences are reduced when multimodal context is considered. Specifically, the response profile of abstract words becomes more concrete-like when these are processed in visual scenes highly related to their meaning. Conversely, when the visual context is unrelated to a given concrete word, the activation pattern resembles more that of abstract conceptual processing. These results suggest that while concepts generally encode habitual experiences, the underlying neurobiological organisation is not fixed but depends dynamically on available contextual information.

## Introduction

Humans acquire, and process language in situated multimodal contexts, through dynamic interactions with their environment. For example, children may learn what the word 'tiger' means primarily via sensory-motor experience: they see one on TV, or they are told that a tiger looks like a big cat. Conversely, the experience required for understanding the more abstract concept of 'good' will likely include an evaluation of the rational and emotional motives underscoring intentional actions. Consequently, while more concrete concepts have external physical references (they refer to objects or actions that are easily perceived in the world), more abstract concepts do not necessarily have such references (they generally refer more to cultural and societal constructs or peoples' inner states of mind) (*Villani et al., 2019*). Is this difference reflected in concrete and abstract representations in the brain during naturalistic processing? And are these static or can they change as a function of the multimodal contexts in which processing occurs?

**eLife digest** When we learn and use language, we deal with two main types of concepts. Concrete concepts, which refer to things we directly experience (like a chair, running or the colour blue), and abstract concepts, which refer to ideas that we are unable to sense directly (like truth, democracy or love).

Most studies have looked at how people process these concepts in isolation, such as by reading single words on a screen. This revealed that the human brain processes each concept differently, with concrete concepts typically activating brain regions involved in sensory and motor experiences, and abstract concepts activating regions involved in emotion and complex thinking.

However, the experiments conducted in these studies do not represent real life situations, where humans often encounter and process both concepts simultaneously. For instance, at the same time as processing language, someone may also be seeing, hearing, and experiencing other things in their environment.

Kewenig et al. wanted to understand whether the brain processes abstract and concrete concepts differently depending on what a person may be visualizing at the same time. To achieve this, they used a technique known as functional MRI to record which regions of the brain are activated as participants watched different movies.

The team found that when abstract concepts (such as love) appeared with related visual information (such as people kissing), the brain processed them more like concrete concepts, engaging sensory and motor regions. Conversely, when concrete concepts (like a chair) appeared without related visual information, the brain processed them more like abstract concepts, engaging regions involved in complex thinking. This suggests that the way the human brain processes meaning is very dynamic and constantly adapting to available contextual information.

These findings could help improve artificial intelligence systems that process language and visual information together, making them better at understanding context-dependent meaning. They might also benefit people with language disorders by informing the development of more effective therapies that consider how context affects understanding. However, more research is needed to confirm these findings and develop practical applications, particularly studies testing whether similar brain patterns occur in other natural situations.

Most studies of concrete and abstract processing are not naturalistic in that they present words or sentences isolated from the rich contexts in which we usually process them. Collectively, these studies suggest that concrete and abstract concepts engage separate brain regions involved in processing different types of information. (*Bedny and Thompson-Schill, 2006*; *Binder et al., 2009*; *Sabsevitz et al., 2005*; *Wang et al., 2010*; *Vigliocco et al., 2014*). Concrete words engage regions involved in experiential processing (*Barsalou et al., 2003*). For example, motor-related cortices activate during the processing of action verbs like 'throw' (*Hauk et al., 2004*), or action-related nouns like 'hammer' (*Vigliocco et al., 2006*; *Kiefer and Pulvermüller, 2012*), auditory cortices for sound-related words like 'telephone' (*Goldberg et al., 2006*; *Kiefer et al., 2008*), and visual cortices for color-related words like 'yellow' (*Hsu et al., 2011*; *Simmons et al., 2007*). These results are consistent with the view that we learn and neurobiologically encode concrete concepts in terms of the sensory and motor experiences associated with their referents.

In contrast, some studies of abstract concepts have found greater activation in brain regions associated with general linguistic processing (*Binder et al., 2005*; *Mellet et al., 1998*; *Noppeney et al., 2004*; *Sabsevitz et al., 2005*). These findings suggest that abstract concepts are learned by understanding their role in a linguistic context, including semantic relationships with other words (e.g. 'democracy' is understood through its relationships to words like 'people,' 'parliament,' 'politics,' etc., e.g. *Jones et al., 2012*). However, neurobiological data also support the view that subcategories of abstract concepts retain sensorimotor information (*Harpaintner et al., 2022*; *Harpaintner et al., 2020*; *Harpaintner et al., 2018*; *Fernandino et al., 2022*) as well as social information (*Villani et al., 2019*; *Conca et al., 2021a*) and internal/interoceptive/affective experiences (*Oosterwijk et al., 2015*; *Vigliocco et al., 2014*), which are also important for learning abstract concepts (*Ponari et al., 2018*). Thus, abstract concepts constitute a more heterogeneous category (*Roversi et al.,*

**Table 1.** Complete Meta-analytic description of clusters.

| Dimension | Abstract Clusters (N=35) | Concrete Clusters (N=20) | Kruskal-Wallis Test |
|---|---|---|---|
| Autob. Memory | 9 | 1 | H(2)=4, P=0.05 |
| Valence | 8 | 0 | H(2)=5.6, P=.01 |
| Theory of Mind | 6 | 0 | H(2)=4.8, P=0.03 |
| Nausea | 6 | 0 | H(2)=4.8, P=0.03 |
| Pain | 5 | 0 | H(2)=4, P=0.05 |
| Movement | 2 | 11 | H(2)=12.4, P<.001 |
| Social/Empathy | 4 | 1 | H(2)=0.7, P=0.42 |
| Touch | 3 | 0 | H(2)=1.8, P=0.18 |
| Speech | 5 | 6 | H(2)=1.1, P=0.29 |
| Language | 8 | 4 | H(2)=0.08, P=0.78 |
| Reading | 3 | 0 | H(2)=1.8, P=0.18 |
| Reward/Motivation | 7 | 3 | H(2)=0.24, P=0.63 |
| Vision | 0 | 3 | H(2)=5.3, P=0.02 |
| Listening | 3 | 2 | H(2)=0.02, P=0.88 |
| Planning | 1 | 0 | H(2)=1.7, P=0.19 |
| Calculation | 1 | 0 | H(2)=1.7, P=0.19 |

*2013*; *Zdrazilova et al., 2018*; *Villani et al., 2019*; *Muraki et al., 2020*; *Muraki et al., 2020*; *Kiefer et al., 2022*).

A limitation of these studies is that they have only investigated the processing of decontextualized concepts (see e.g. *Table 1* in a recent review by *Del Maschio et al., 2022*). That is, they (often implicitly) assume that conceptual representations in the brain are the product of a stable set of regions processing different types of information depending on whether a concept is concrete or abstract. However, this dichotomy may not account for the way in which we typically process concepts (*Lebois et al., 2015*), given that the information encoded during conceptual processing depends on the contextual information available. For example, the situated (i.e. contextualized in the discourse but also in the physical setting in which processing occurs) processing of concrete concepts like 'chair' could be linked to many abstract internal elements like goals ('I want to rest'), motivations ('I have been standing for 2 hr'), emotions ('I would like to feel comfortable'), and theory of mind ('is that older person more in the need of this chair than me?'). Conversely, an abstract concept like 'truth' is no longer particularly abstract when used in reference to a perceived physical situation (such as 'snowing') that matches the utterance's meaning ('it is true that it is snowing'). Here, 'truth' refers to a concrete state of the world (*Barsalou et al., 2018*).

Indeed, previous work has postulated flexible conceptual processing in experiential brain circuits (*Binder and Desai, 2011*; *Pulvermüller, 2018a*). Behavioral data support the view that contextual information can affect conceptual processing (e.g., *Chambers et al., 2004*; *Cooper, 1974*; *Tanenhaus et al., 1995*). For example, when an object is depicted in a visual context consistent with its use, the action associated with using the object is more readily available than when the context is more consistent with picking the object up (*Kalénine et al., 2014*). There is also neurobiological evidence that objects visually present in a situation can influence conceptual processing (*Hoffman et al., 2013*; *Yee and Thompson-Schill, 2016*). For example, task-related color-congruency of objects correlates with less activation of brain regions involved in color perception during processing – likely because less retrieval of detailed color knowledge was necessary (*Hsu et al., 2011*). Dynamic, context-dependent recruitment of visual and motor-related areas during semantic processing has also been established (*Hoenig et al., 2008*; *van Dam et al., 2012*; *Popp et al., 2019*). An understanding of conceptual

knowledge as static and context-independent is insufficient to account for these dynamics (*Pulvermüller, 2018b*).

However, no previous study has addressed whether the brain areas associated with concrete and abstract concepts are fixed or recruited in a more dynamic way during semantic processing. The present study aims to fill this gap and test the following two predictions. First, we submit that results from previous investigations of conceptual processing, which generally depict a stable dichotomy between concrete and abstract words, reflect the average experiential information of the type of situational context in which concepts are habitually experienced. Therefore, we predict that the neurobiological representation of concrete concepts, will be related to associated brain regions, because they retain experiences related to their physical referents that are predominantly characterized by sensory and motor information, (*Pulvermüller, 2018b*; *Willems et al., 2010*). In contrast, because their representations mostly reflect information related to internal/interoceptive/affective experience as well as linguistic information, we expect abstract concepts to activate brain regions associated with emotional, interoceptive, and general linguistic processing (*Reinboth and Farkaš, 2022*).

Second, the reviewed work also suggests that these habitual representations are not necessarily stable and might change during naturalistic processing depending on the specific contextual information available. We specify two context conditions: a concept is *displaced* if its context offers little or no visual information related to the concept's external sensory-motor features. In contrast, a concept is *situated*, if its visual context contains objects related to its meaning. We predict that when a concrete concept is processed in displaced situations (e.g. 'cat' processed when discussing the general character traits of cats vs dogs), response profiles will shift towards more internalized processing shared with abstract concepts. In contrast, when an abstract concept is processed in a situated context (for example the word 'love' processed in a scene with people kissing), its representation will more heavily draw on regions involved in processing external, visual information that otherwise characterize more concrete concepts.

We propose that this erosion of the concrete/abstract dichotomy for contextualized processing shows that both concrete and abstract concepts draw on information related to experience (external and internal) as well as linguistic association. Which associated neurobiological structures are engaged during processing depends dynamically on the contextual information available. This way of thinking about the representational nature of conceptual knowledge may help reconcile contradictory evidence concerning the involvement of different brain areas during conceptual processing (for example as discussed in *Patterson et al., 2007*; *Hoffman et al., 2018*).

## Results
### Conceptual processing across contexts

Consistent with previous studies, we predicted that across naturalistic contexts, concrete and abstract concepts are processed in a separable set of brain regions. To test this, we contrasted concrete and abstract modulators at each time point of the IRF (*Figure 1*). This showed that concrete produced more modulation than abstract processing in parts of the frontal lobes, including the right posterior inferior frontal gyrus (IFG) and the precentral sulcus (*Figure 1, red*). Known for its role in language processing and semantic retrieval, the IFG has been hypothesized to be involved in the processing of action-related words and sentences, supporting both semantic decision tasks and the retrieval of lexical semantic information (*Bookheimer, 2002*; *Hagoort, 2005*). The precentral sulcus is similarly linked to the processing of action verbs and motor-related words (*Pulvermüller, 2005*). In the temporal lobes, greater modulation occurred in the bilateral transverse temporal gyrus and sulcus, planum polare, and temporale. These areas, including primary and secondary auditory cortices, are crucial for phonological and auditory processing, with implications for the processing of sound-related words and environmental sounds (*Binder and Desai, 2011*) . The superior temporal gyrus (STG) and sulcus (STS) also showed greater modulation for concrete words and these are said to be central to auditory processing and the integration of phonological, syntactic, and semantic information, with a particular role in processing meaningful speech and narratives (*Hickok and Poeppel, 2007*) . In the parietal and occipital lobes, more concrete modulated activity was found bilaterally in the precuneus, which has been associated with visuospatial imagery, episodic memory retrieval, and self-processing operations and has been said to contribute to the visualization aspects of concrete concepts (*Cavanna and Trimble,*

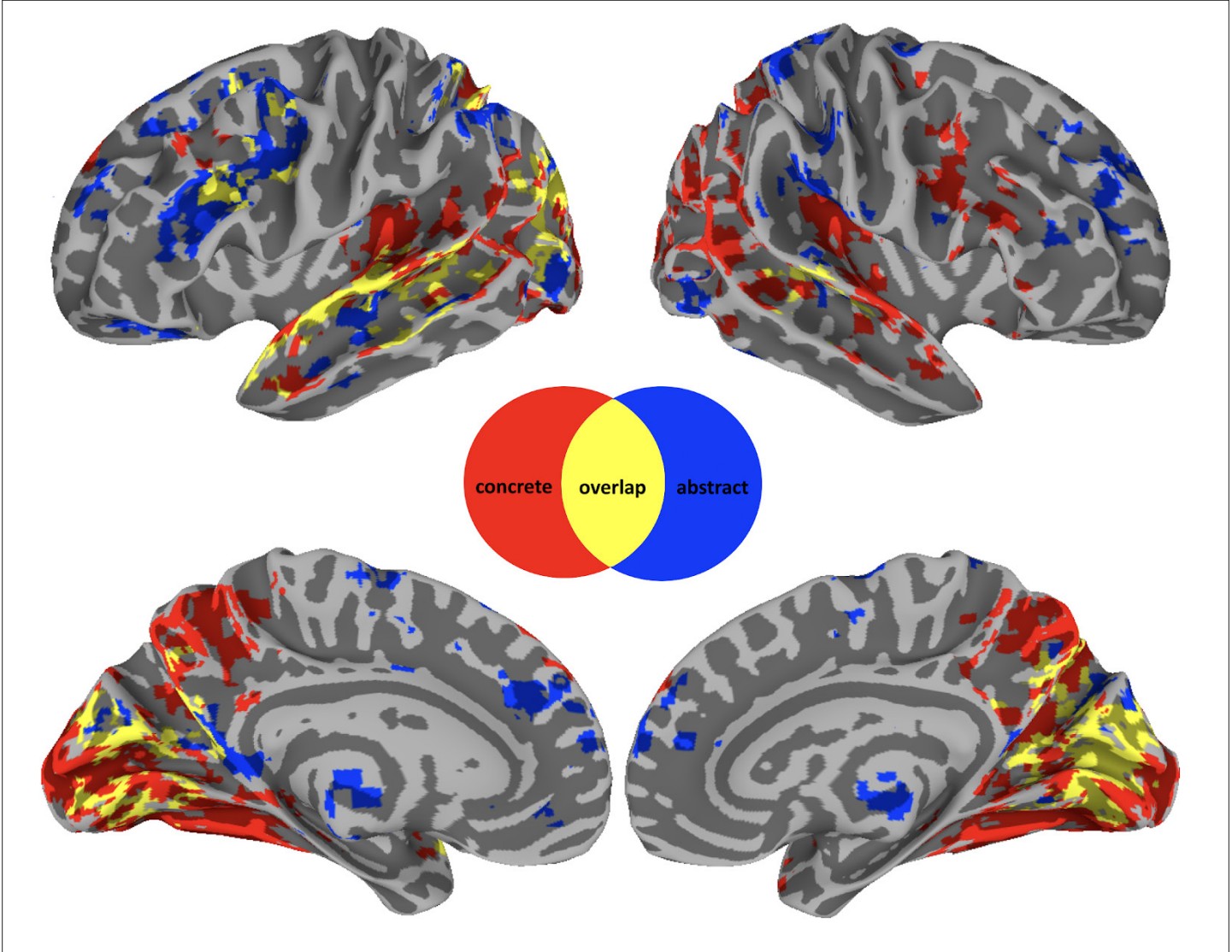

**Figure 1.** Neurobiology of conceptual processing across contexts. Colored regions show group-level results from a linear mixed effect model and subsequent general linear tests contrasting activity for concrete (red) versus abstract (blue) modulation at each of 20 timepoints after word onset. Overlapping regions (yellow) indicate a concrete and abstract difference at one of these timepoints. Results are thresholded and corrected for multiple comparisons at α=0.01 and displayed with a cluster size ≥ 20 voxels.

The online version of this article includes the following figure supplement(s) for figure 1:

**Figure supplement 1.** Comparison between overlap and language regions.

**Figure supplement 2.** Time course of activation.

*2006*) . More activation was also found in large swaths of the occipital cortices (running into the inferior temporal lobe), and the ventral visual stream. These regions are integral to visual processing, with the ventral stream (including areas like the fusiform gyrus) particularly involved in object recognition and categorization, linking directly to the visual representation of concrete concepts (*Simmons et al., 2007*). Finally, subcortically, the dorsal and posterior medial cerebellum were more active bilaterally for concrete modulation. Traditionally associated with motor function, some studies also implicate the cerebellum in cognitive and linguistic processing, including the modulation of language and semantic processing through its connections with cerebral cortical areas (*Stoodley and Schmahmann, 2009*).

Conversely, activation for abstract was greater than concrete words in the following regions (*Figure 1*, blue): In the frontal lobes, this included the right anterior cingulate gyrus, lateral and medial aspects of the superior frontal gyrus. Being involved in cognitive control, decision-making, and

emotional processing, these areas may contribute to abstract conceptualization by integrating affective and cognitive components (*Shenhav et al., 2013*) . More left frontal activity was found in both lateral and medial prefrontal cortices, and in the orbital gyrus, regions which are key to social cognition, valuation, and decision-making, all domains rich in abstract concepts (*Amodio and Frith, 2006*). In the parietal lobes, bilateral activity was greater in the angular gyri (AG) and inferior parietal lobules, including the postcentral gyrus. Central to the default mode network, these regions are implicated in a wide range of complex cognitive functions, including semantic processing, abstract thinking, and integrating sensory information with autobiographical memory (*Seghier, 2013*). In the temporal lobes, activity was restricted to the STS bilaterally, which plays a critical role in the perception of intentionality and social interactions, essential for understanding abstract social concepts (*Frith and Frith, 2003*). Subcortically, activity was greater, bilaterally, in the anterior thalamus, nucleus accumbens, and left amygdala for abstract modulation. These areas are involved in motivation, reward processing, and the integration of emotional information with memory, relevant for abstract concepts related to emotions and social relations (*Haber and Knutson, 2010*; *Phelps and LeDoux, 2005*).

Finally, there was an overlap in activity between modulation of both concreteness and abstractness (*Figure 1*, yellow). The overlap activity is due to the fact that we performed general linear tests for the abstract/concrete contrast at each of the 20 timepoints in our group analysis. Consequently, overlap means that activation in these regions is modulated by both concrete and abstract word processing but at different time-scales. In particular, we find that activity modulation associated with abstractness is generally processed over a longer time-frame (for a comparison of significant timing differences see *Figure 1—figure supplement 2*). In the frontal, parietal, and temporal lobes, this was primarily in the left IFG, AG, and STG, respectively. Left IFG is prominently involved in semantic processing, particularly in tasks requiring semantic selection and retrieval, and has been shown to play a critical role in accessing semantic memory and resolving semantic ambiguities, processes that are inherently time-consuming and reflective of the extended processing time for abstract concepts (*Thompson-Schill et al., 1999*; *Wagner et al., 2001*; *Hoffman et al., 2015*). The STG, particularly its posterior portion, is critical for the comprehension of complex linguistic structures, including narrative and discourse processing. The processing of abstract concepts often necessitates the integration of contextual cues and inferential processing, tasks that engage the STG and may extend the temporal dynamics of semantic processing (*Ferstl et al., 2008*; *Vandenberghe et al., 2002*). In the occipital lobe, processing overlapped bilaterally around the calcarine sulcus, which is associated with primary visual processing (*Kanwisher et al., 1997*; *Kosslyn et al., 2001*).

## Meta-analytic results

Overall, these results suggest that concrete modulation engages sensory and motor regions more, whereas abstract words engage regions more associated with semantic as well as internal/interoceptive/affective processing. Both categories overlap (though necessarily at different time points) in regions typically associated with word processing. However, these interpretations are based on informal reverse inference. To more formally and quantitatively evaluate this distinction between concrete and abstract words, we employed meta-analytic description and reverse correlation analyses. Both test whether brain regions involved in concrete and abstract conceptual processing reflect different types of habitual experience (i.e. sensory-motor vs internal/interoceptive/affective).

Term-based labeling demonstrates that significantly more concrete clusters are related to the term 'Movement' compared to abstract clusters (H(2) = 12.4, p<0.001; *Figure 2, red*). In contrast, abstract clusters are more related to terms that are arguably associated with internal/interoceptive/affective processing compared to concrete activation clusters, i.e., 'Autobiographical Memory,' 'Nausea,' 'Pain,' 'Reward/Motivation,' and 'Valence' (all ps <0.05; *Figure 2*). Finally, 'Language' was the only term more associated with overlapping clusters than either concrete (H(2) = 7, p<0.001) or abstract clusters (H(2) = 4, p=0.045; *Figure 2*). For meta-analytic associations of each individual cluster, see *Table 1*.

## Peaks and valleys results

Comparing dimensions for abstract vs concrete modulated clusters, we found significantly more concrete compared to abstract clusters associated with the dimension 'Torso' H(2)=7, p<0.001. Three concrete clusters were associated with 'Haptic' and 'Mouth,' which was also significantly more than for abstract clusters (all tests H(2)=5.2, all p's = 0.02). Two concrete clusters with 'Foot_Leg' compared to

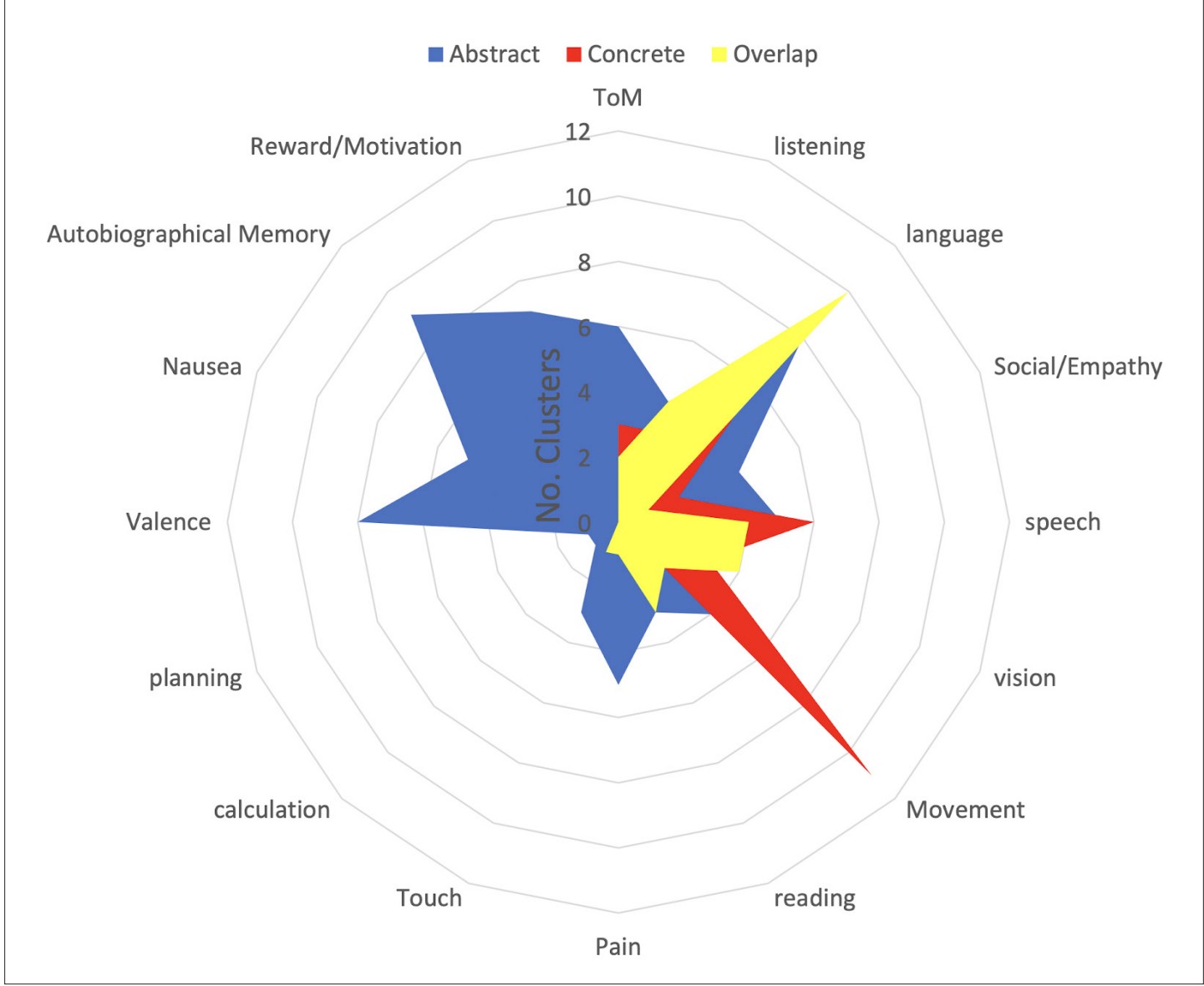

**Figure 2.** Meta-analytic description of conceptual processing across contexts. We used the Neurosynth meta-analysis package to find the terms associated with the centers of mass for each concrete (red), abstract (blue), and overlap (yellow) cluster from *Figure 1*. Numbers refer to the number of activation clusters associated with each meta-analytic term. There were significantly more concrete than abstract clusters for the term 'Movement' (p<0.001), whereas there were more abstract compared to concrete clusters for 'Autobiographical Memory,' 'Nausea,' 'Pain,' 'Theory of Mind,' and 'Valence' (all p's <0.05). The term 'language' was significantly more associated with overlap clusters compared to concrete (p<0.001) and abstract clusters (p=0.045).

0 abstract clusters was not significant, but the mean was in the expected direction H(2)=3.4, p=0.06 All concrete clusters are displayed in *Figure 3* (red). Conversely, eight abstract clusters were significantly more associated with the dimension 'Valence' than concrete clusters (H(2)=8.3, p<0.001). Three abstract clusters were associated with the dimension 'Auditory,' which was not significantly different from concrete clusters (H(2)=1.9, p=0.17). All abstract clusters are displayed in *Figure 3* (blue). Finally, five clusters in which modulation through concreteness and abstractness overlapped (though at different time points) were significantly more associated with the dimension 'Mouth' compared to two concrete clusters H(2) = 5.1, p=0.03 and 0 abstract clusters H(2)=7.8, p<0.001. All overlap clusters are displayed in *Figure 3* (yellow). For all results of the peak and valley tests for each individual cluster, see *Table 2*.

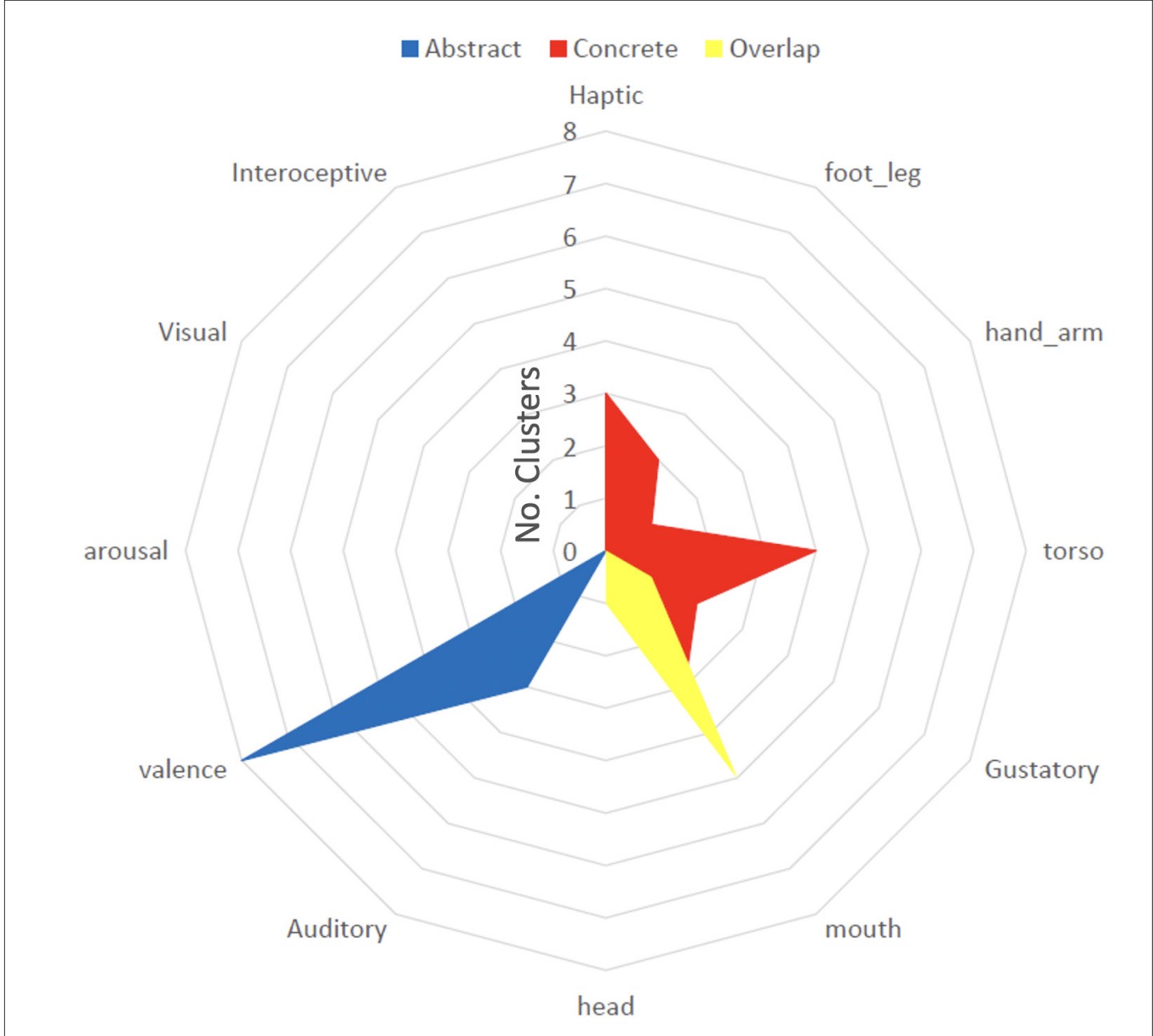

**Figure 3.** Peak and valley analysis results for understanding conceptual processing across contexts. We extract the type of information processed in each activation cluster by looking at experience-based features of movie words that are aligned with significantly more peaks than valleys (see *Figure 1*). Words highly rated on the sensorimotor dimensions 'Haptic,' 'Hand_Arm,' and 'Torso' were significantly more associated with concrete clusters (red, all p's <0.05), 'Valence' with abstract clusters (blue, p<0.001) and 'Mouth' with overlap clusters (yellow, p's <0.05). For some features/terms, there were never significantly more words highly rated on that dimension occurring at peaks compared to valleys, so they do not have any significant clusters.

The online version of this article includes the following figure supplement(s) for figure 3:

**Figure supplement 1.** Peak and valley analysis for a 4 s lag.

**Figure supplement 2.** Peak and valley analysis at the individual participant level (5s lag).

**Figure supplement 3.** Peak and valley analysis after averaging results for individual participants for abstract clusters only.

**Figure supplement 4.** Peak and valley analysis after averaging results for individual participants for concrete clusters only.

**Figure supplement 5.** Peak and valley analysis after averaging results for individual participants for overlap clusters only.

**Figure supplement 6.** Test of nonlinearity for peak and valley analysis.

**Figure supplement 7.** Overview of the features that showed nonlinear interactions.

**Table 2.** Peak and valley results between concrete and abstract activation clusters.

| Dimension | Abstract clusters (N=35) | Concrete clusters (N=20) | Kruskal-Wallis test |
|---|---|---|---|
| Valence | 9 | 1 | H(2)=4, p=0.05 |
| Interoceptive | 8 | 0 | H(2)=5.6, p=0.01 |
| Arousal | 6 | 0 | H(2)=4.8, p=0.03 |
| Auditory | 6 | 0 | H(2)=4.8, p=0.03 |
| Visual | 5 | 0 | H(2)=4, p=0.05 |
| Head | 2 | 11 | H(2)=12.4, p<0.001 |
| Haptic | 0 | 3 | H(2)=5.3, p=0.02 |
| Foot_Leg | 0 | 2 | H(2)=3.5, p=0.06 |
| Hand_Arm | 0 | 1 | H(2)=1.7, p=0.19 |
| Torso | 0 | 4 | H(2)=7.3, p=0.01 |
| Gustatory | 0 | 2 | H(2)=3.5, p=0.06 |
| Mouth | 0 | 3 | H(2)=5.3, p=0.02 |
| Head | 0 | 0 | / |

## Conceptual processing in context

Activation associated with the main effect of word_type overlapped with processing of concrete and abstract words across context in superior temporal sulcus, superior temporal gyrus and middle temporal gyrus (bilateral), in angular gyrus (bilateral), in the central sulcus and precentral and post-central gyrus (right hemisphere), in lateral and medial frontal cortices as well as in the occipital lobe (see *Figure 4A*). Activation for the main effect of context was found bilaterally in posterior temporal lobe at the intersection with occipital lobe, as well as in nodes of the default mode network (DMN), including precuneus, medial prefrontal regions and angular gyrus (*Figure 4B*). The interaction between word_type and context-modulated activity in the main nodes of the DMN (amongst other regions), including precuneus, medial prefrontal regions, and angular gyrus (all bilaterally, see *Figure 4C*). Indeed, the thresholded interaction map with 1501 voxels was 'decoded' using the Neurosynth package, where the Pearson correlation is computed between the vectorized map and all the maps in the Neurosynth database. The top four associated terms (excluding brain regions or methodological terms) were the 'DMN' (r(1500)=0.194, p<0.001), 'Default Mode' (r(1500)=0.219, p<0.001), and 'Default' (r(1500)=0.226, p<0.001) as well as 'Semantic Control' (r(1500)=0.206, p<0.001).

To better understand the nature of this interaction and how it relates to response profiles associated with concreteness and abstractness across contexts, we contrasted concrete vs abstract modulation in situated and displaced conditions (collapsing across timepoints, as we had no prediction about timing differences). This comparison is displayed in *Figure 5*. We then extracted the resulting activation maps as masks and spatially correlated them with the brain mask obtained from the activation map contrasting concreteness and abstractness across contexts. This comparison is displayed in *Figure 6*.

To quantify this comparison, we used cosine similarity to calculate spatial correlation measures between the unthresholded results from contrasting concrete and abstract modulations in displaced and situated context with the unthresholded contrasts between concrete and abstract modulations across contexts (*Figure 1*, red is the thresholded version of this map). This shows that in situated context, the contrasted modulation by abstractness overlaps more with concreteness across context (r(72964)=0.64, p<0.001) compared to displaced concreteness (r(72964)=0165, p=0.476). Concreteness across contexts overlaps with situated abstractness (*Figure 5A*, red) bilaterally in the fusiform, occipital lobe, inferior and superior parietal lobules and with displaced concreteness bilaterally in the occipital lobe, as well as large swaths of the superior temporal gyrus (*Figure 6*).

Conversely, in displaced context, the contrasted modulation by concreteness (*Figure 5B*, blue) overlaps more with the pattern of activity modulated by abstractness across context (r(72,964) = 0.49, p<0.001) compared to situated abstractness (r(72,964) = 0.21, p<0.001). Abstractness across

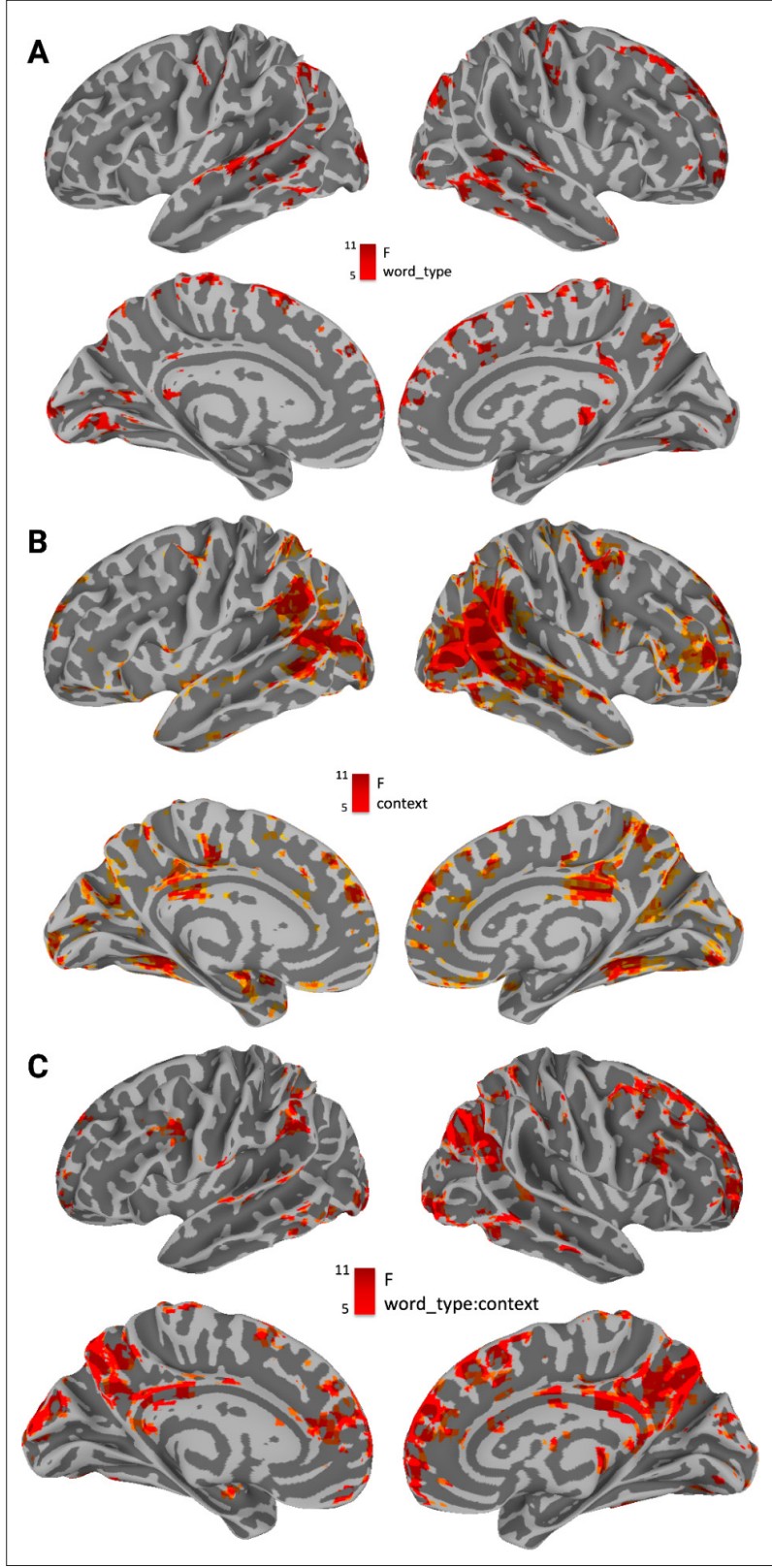

**Figure 4.** Main effects of word_type, context, and their interaction. (A) Main effect of word-type. Most significantly modulated areas include superior temporal sulcus, superior temporal gyrus and middle temporal gyrus (bilateral), angular gyrus (bilateral), the central sulcus and precentral and postcentral gyrus (right hemisphere), as well as lateral and medial frontal cortices and the occipital lobe. (B) Main effect of context. Most significantly modulated

*Figure 4 continued on next page*

*Figure 4 continued*

areas include the intersection between posterior temporal and occipital lobe, the Precuneus, Middle Prefrontal Cortex, as well as Angular Gyrus, and right inferior frontal gyrus. (**C**) Interaction between context (high/low) and word type (abstract/concrete). Most significantly modulated areas include the Precuneus, Middle Prefrontal Cortex as well as Middle Frontal Gyrus, Angular Gyrus, and Posterior Cingulate Cortex. These correspond to the nodes of the default mode network, as well as areas commonly associated with semantic control. This was confirmed by using the neurosynth decoder on the unthresholded brain image - top keywords were 'Semantic Control' and 'DMN.' All displayed results are thresholded and corrected for multiple comparisons at α=0.01 and displayed with a cluster size ≥ 20 voxels.

contexts overlaps with displaced concreteness bilaterally in large portions of the inferior parietal lobule, including supramarginal gyrus and post superior temporal sulcus, up to the intersection of the occipital and temporal lobes, lingual gyrus in particular (*Figure 6*). Overlap activation could also be found bilaterally in anterior thalamus and medial prefrontal regions.

## Discussion

Conceptual processing is typically investigated in experiments where words are stripped away from their naturally occurring context: most studies use isolated words, and sometimes sentences (see *Table 1* in *Del Maschio et al., 2022*). However, conceptual processing in its ecology occurs in rich multimodal contexts. Our study investigated naturalistic conceptual processing during movie-watching to begin to understand the effect of multimodal context on the neurobiological organization of real-world conceptual representation.

### Conceptual processing across contexts

First, we asked where in the brain concrete and abstract concepts are processed across different contexts as well as the type of information they encode. Given the hypothesis that conceptual representations reflect contextual information, we expected a set of regions that correspond to the most typical set of experiences (e.g. as encountered during word learning in development) to activate across different contexts. Specifically, we expected concrete conceptual encoding to activate regions more involved in sensory and motor processing and abstract conceptual encoding to activate regions associated with more internal/interoceptive/affective as well as general linguistic processing (*Anderson et al., 2019*; *Meteyard et al., 2012*; *Binder et al., 2005*).

Indeed, we found a general tendency for concrete and abstract words to activate regions associated with different experiences (*Figure 1*). Consistent with prior work, concrete words were associated with multiple regions involved in sensory and motor processing (*Mkrtychian et al., 2019*), including most of the visual system (*Gao et al., 2019*) and the right frontal motor system (*Pulvermüller, 2005*). In contrast, abstract words engaged regions typically associated with internal/interoceptive/affective processing (anterior thalamus, somatosensory cortex)(*Harpaintner et al., 2018*; *Villani et al., 2021*), autobiographical memory (anterior medial prefrontal regions) (*Conca et al., 2021a*), and emotional processing and regulation (anterior medial prefrontal regions, orbital prefrontal cortex, dorsolateral prefrontal cortex, nucleus accumbens and amygdala) (*Vigliocco et al., 2014*; *Conca et al., 2021b*).

Consistent with this, both meta-analytic and peak and valley analyses showed that concrete regions were more associated with sensory-motor properties (e.g. 'Movement' and 'Hand_Arm') whereas abstract regions were more associated with internal/interoceptive/affective properties (e.g. 'Valence;' *Figures 2 and 3*). Together, these results provide evidence from naturalistic processing that concrete and abstract concepts encode different types of experiences (*Vigliocco et al., 2009*; *Barsalou and Wiemer-Hastings, 2005*; *Kiehl et al., 1999*).

At the level of brain regions, our study aligns with previous literature identifying distinct brain regions engaged in processing abstract versus concrete words. Specifically, our results show greater activation for concrete words in temporo-parieto-occipital regions. These areas include the bilateral middle temporal gyrus, the left fusiform gyrus, and the bilateral angular gyrus, among others. Conversely, our study found that abstract word processing preferentially engages a network of regions within the left hemisphere, including the inferior frontal gyrus (IFG), superior and middle temporal gyri, and the inferior parietal lobule.

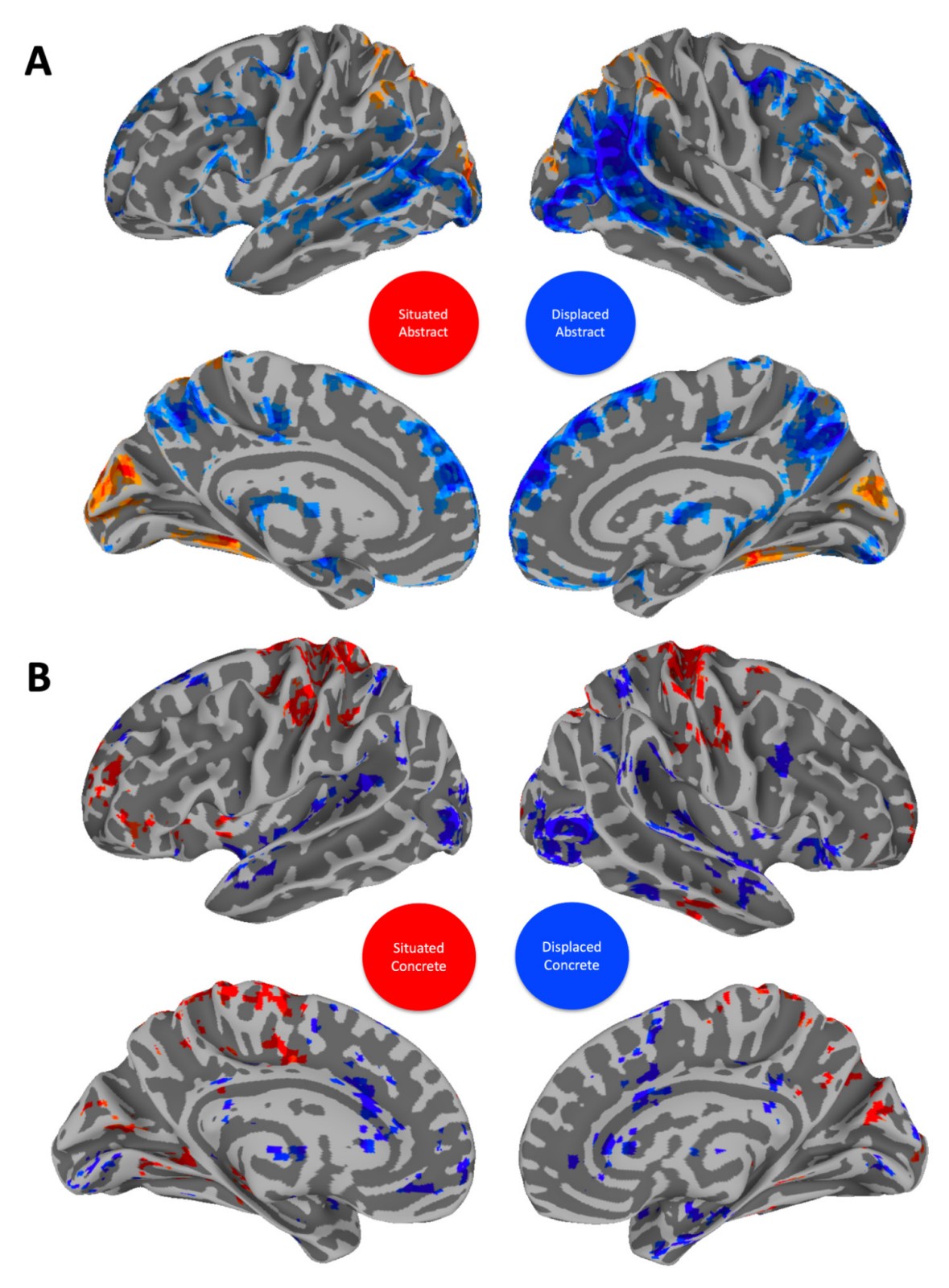

**Figure 5.** Contrasts between situated abstract and displaced abstract (**A**) as well as situated concrete and displaced concrete (**B**). The displaced concrete activation mask was later correlated with abstract processing across context (see *Figure 6*). The situated abstract activation mask was later correlated with concrete processing across context (see *Figure 6*). Nodes of the default mode network (DMN) are especially active in the displaced condition for both abstract and concrete words. Visual and sensorimotor areas are especially active in situated conditions for both abstract and concrete words. Results are thresholded and corrected for multiple comparisons at α=0.01 and displayed with a cluster size ≥ 20 voxels.

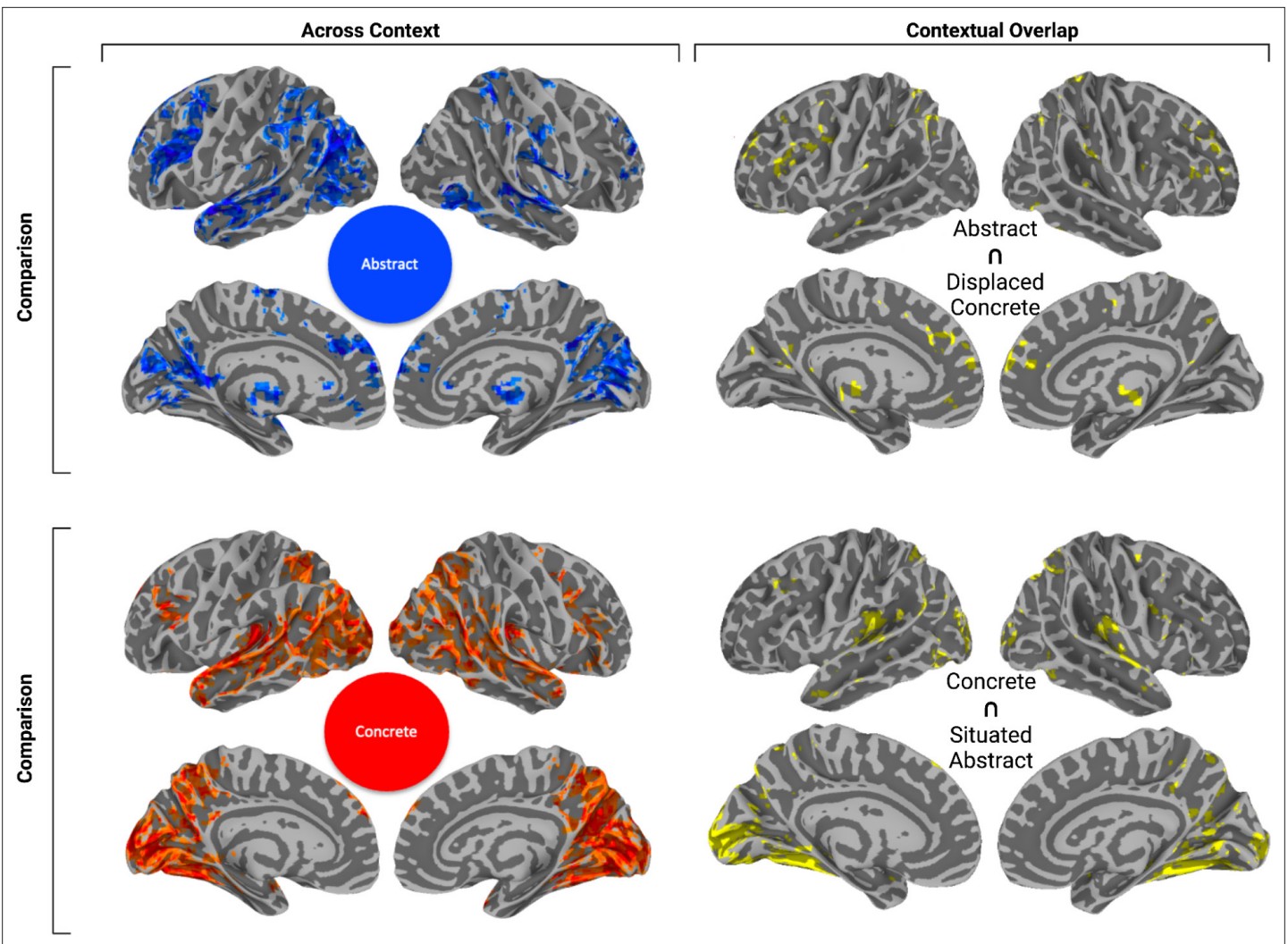

**Figure 6.** Spatial overlap between thresholded statistical brain images of concrete and abstract conceptual processing obtained from the original analysis across contexts situatedness/displacement contrasts (on the left). Original brain maps of our analysis across context are split into abstract (top) and concrete (bottom) on the right. The overlap between displaced concrete and abstract was (r(72,964) = 0.49, p<0.001), the overlap between situated abstract and concrete was (r(72,964)=0.64, p<0.001). All maps were thresholded at α=0.01 with a cluster size ≧ 20 voxels.

However, the regions involved in processing concrete and abstract concepts across contexts did not imply a fully dichotomous encoding of experiences. First, we found that regions involved in sensory (mostly in visual cortices) and motor processing are involved in processing both types of words (*Figure 1*). Moreover, we found overlap activation in regions associated with language processing in general (*Tang et al., 2022*, *Figure 1*). Such results are in line with proposals in which both concrete and abstract representations rely on experiential information as well as their linguistic relationships with other words (e.g. *Vigliocco et al., 2009*; *Vigliocco et al., 2009*; *Piantadosi and Hill, 2022*). This latter hypothesis is also supported by our Peaks and Valleys analysis, more specifically that information related to 'Mouth' (i.e. the language organ) drives activation in overlap clusters. This is furthermore evidence against hypotheses in which the mouth is specifically associated with abstract concepts (*Borghi and Zarcone, 2016*).

## Conceptual processing in context

Though results across contexts presumably represent a form of experiential central tendency, the behavioral, neuroimaging, and electrophysiological literature suggests that conceptual representations might not be stable and may vary as a function of context (*Elman, 1995*; *Spivey and Dale, 2006*; *Cai and Vigliocco, 2018*; *Kutas and Federmeier, 2011*; *Yee and Thompson-Schill, 2016*; *Deniz*

*et al., 2023*). For this reason, we conducted a second set of analyses with the goal of understanding the extent to which representations associated with concrete or abstract conceptual processing in the brain change as a function of context (*Barsalou et al., 2018*).

We find that brain activation underlying concrete and abstract conceptual processing fundamentally changes as a function of visual context. We compared the activation profiles of concrete and abstract concepts in displaced and situated contexts with the activations obtained when collapsing across contexts. Our results show that concrete concepts become more abstract-like in displaced contexts with less relevant visual information (*Figure 4B*). Overlap between activation for concrete concepts in displaced conditions and abstract concepts across context (*Figure 5A*) can be found in ACC, thalamus, and large swaths of the anterior, middle, and posterior temporal lobe. We propose that this is because, when a concrete concept is processed in a displaced context, its representation will relate more to internal/interoceptive variables and linguistic associations, which are usually encoded by abstract concepts. Conversely, abstract concepts become more concrete-like when they are highly situated (*Figure 4A*). Overlap between activation for abstract concepts in situated conditions and concrete concepts across context (*Figure 5B*) can be found in fusiform and the occipital lobe (bilateral). We propose that this is because an abstract concept processed in a situated context relates more to external visual information, which is usually encoded by concrete concepts. A consequence of this finding is that the concrete/abstract distinction is neurobiologically less stable than might be assumed. Brain regions 'switch alliance' during concrete or abstract word processing depending on context.

What is the neurobiological mechanism behind contextual modulation of conceptual encoding in the brain? Our results indicate that variance in visual context interacted with word-type (both concrete and abstract) in regions commonly defined as the DMN, as well as a set of prefrontal regions associated with semantic control (*Ralph et al., 2017*; *Hoffman et al., 2018*; *Figure 3—figure supplements 1 and 2*). Recent literature on the role of the DMN suggests that these regions reactivate memories (*Crittenden et al., 2015*; *Konishi et al., 2015*; *Murphy et al., 2018*; *Sormaz et al., 2018*; *Yee and Thompson-Schill, 2016*; *Andrews-Hanna et al., 2014*, *Vatansever et al., 2017*) and contexts-specific information (*Hahamy et al., 2023*), possibly to form contextually relevant situation models (*Chen et al., 2015*; *Raykov et al., 2020*, *Ranganath and Ritchey, 2012*, *Smith et al., 2021*, *Smith et al., 2017*), in order to guide semantic cognition (*Binder et al., 1999*; *Fernandino et al., 2016*; *Yeshurun et al., 2021*; *Tong et al., 2022*).

Breaking up the interaction between word_type and context, we find that the DMN is especially involved in displaced conditions for both concrete and abstract conceptual processing (see *Figure 4A, B* (blue)). These results fit well with evidence suggesting that the DMN supports conceptual processing especially when displaced from sensorimotor input (*Murphy et al., 2018*; *Lanzoni et al., 2020*; *Wang et al., 2010*). Accordingly, the DMN is most strongly activated in the displaced conditions involving abstract concepts. Given their inherent lack of sensorimotor information, abstract concepts offer a greater degree of displacement than their concrete counterparts, thereby demanding a higher engagement of the DMN in these conditions.

In considering the impact of visual context on the neural encoding of concepts generally, it is furthermore essential to recognize that the mechanisms observed may extend beyond visual processing to encompass more general sensory processing mechanisms. The human brain is adept at integrating information across sensory modalities to form coherent conceptual representations, a process that is critical for navigating the multimodal nature of real-world experiences (*Barsalou et al., 2018*; *Smith, 2007*). While our findings highlight the role of visual context in modulating the neural representation of abstract and concrete words, similar effects may be observed in contexts that engage other sensory modalities. For instance, auditory contexts that provide relevant sound cues for certain concepts could potentially influence their neural representation in a manner akin to the visual contexts examined in this study. Future research could explore how different sensory contexts, individually or in combination, contribute to the dynamic neural encoding of concepts, further elucidating the multimodal foundation of semantic processing.

## Conceptual processing and language

The exact relationship between concepts and language remains an open question, but it is undisputed that, as determinants of meaning, concepts are necessary for language (*Jackendoff, 2002*;

*Bloom, 2000*; *Fauconnier and Turner, 2002*). The present study examined language-driven conceptual processing, as we looked at brain activation during word processing. Our results imply that the underlying neurobiological processes are dynamically distributed and contextually determined. This view fits well with models of 'natural' organization of language in the brain where it is argued that language processing more generally is a whole brain process whose patterns of activation are determined by available context (*Skipper and Willems, 2015*; *Skipper and Willems, 2015*). These more distributed regions may be averaged away when indiscriminately analyzed together and following thresholding because (i) they are more variable given they are associated with different experiences (as we have seen here), linguistic categories (e.g. 'formulaic speech;' see *Skipper et al., 2022*), and processes (e.g. different types of syntax) (ii) there are individual differences in all of these (e.g. *Skipper et al., 2022*; *Skipper and Willems, 2015*). These suppositions are supported by the fact that concrete and abstract modulation overlaps in typical perisylvian 'language regions' (*Figure 1—figure supplement 1*).

## Conclusions

Our work emphasizes the merits of investigating conceptual processing in naturalistic multimodal contexts. This paves the way for future analyses systematically quantifying different types of contexts (e.g. in terms of related objects, actions, emotions, or social interactions) and examining how these can affect conceptual processing in the brain. Such work might further our understanding of the neurobiology of conceptual processing in naturalistic settings by clarifying what type of contexts affect processing and how. This may inform the recent development of multimodal large language models, where processing depends on context beyond purely text-based information (*Driess et al., 2023*) - especially in naturalistic settings (*Kewenig et al., 2023*). Apart from commercial applications, gaining a better understanding of the mechanisms underlying naturalistic conceptual processing in the brain might bear important implications for clinical domains, e.g., by informing progress towards helping patients who lost the ability to speak by real-time semantic reconstruction of non-invasive brain recordings with the help of large language models (*Tang et al., 2022*).

# Materials and methods

The present study analyzed the 'Naturalistic Neuroimaging Database (NNDb)' (*Aliko et al., 2020*). All code is made available on a designated repository under (https://github.com/ViktorKewenig/Naturalistic_Encoding_Concepts, copy archived at *ViktorKewenig, 2024*).

## Participants and task

The Naturalistic Neuroimaging Database (*Aliko et al., 2020*, https://openneuro.org/datasets/ds002837/versions/2.0.0) includes 86 right-handed participants (42 females, range of age 18–58 years, M = 26.81, SD = 10.09 years) undergoing fMRI while watching one of 10 full-length movies selected across a range of genres. All had unimpaired hearing and (corrected) vision. None had any contraindication for magnetic resonance imaging (MRI), history of psychiatric or neurological disorder, or language-related learning disabilities. All participants gave informed consent, and the study was approved by the University College London Ethics Committee (Reference Number 143/003).

## Data acquisition and preprocessing

Functional and anatomical images were obtained using a 1.5T Siemens MAGNETOM Avanto, equipped with a 32-channel head coil. Whole-brain images were captured, each consisting of 40 slices per volume at an isotropic resolution of 3.2 mm. These were obtained using a multiband echo-planar imaging (EPI) sequence with no in-plane acceleration, a multiband factor of 4 x, a repetition time of 1 s, an echo time of 54.8 milliseconds, and a flip angle of 75 degrees. Each study participant yielded a number of brain volumes equivalent to movie runtime in seconds. Due to software constraints limiting the EPI sequence to 1 hr of continuous scanning, there were mandatory breaks during the movie for all participants.

The data were preprocessed with AFNI (*Cox, 1996*) and included despiking, slice-time correction, coregistration, blurring, and nonlinear alignment to the MNI152 template brain. The time series underwent smoothing using an isotropic full-width half-maximum of 6 mm, with detrending accomplished

through regressors for motion, white matter, cerebrospinal fluid, and run length. Adjustments were made to account for breaks in movie viewing, and artifacts identified by spatial independent component analysis were regressed out. Detailed information on data acquisition and preprocessing is available in *Aliko et al., 2020* and on openneuro.org.

## Materials

All words in the movies were annotated using automated approaches with a machine learning-based speech-to-text transcription tool from Amazon Web Services (AWS; https://aws.amazon.com/transcribe/). The resulting transcripts contained on and offset timings for individual words. However, as not all words were transcribed or accurately transcribed, timings were subsequently corrected manually.

Concrete and abstract words were selected for the present study from existing *Warriner et al., 2013* norms. In this database, 37,058 words were rated for concreteness on a scale from 0 (not experience-based) to 5 (experience-based) by over 4000 participants. We median split only content words on this scale to yield our set of concrete and abstract words. These were matched for word frequency within 1 SD of mean log frequency (3.61), using the SUTBLEX (US) corpus *Brysbaert and New, 2009*, which contains frequency counts for 72,286 words. Concrete and abstract words were also matched to be within 1SD from mean length measured as number of letters (4.81).

After this matching process, we were left with more concrete words than abstract words in all movies (783 on average for concrete words; 440 for abstract words). To maintain equal numbers in the subsequent analysis, we randomly selected a subset of 440 concrete words in each movie to match the amount of abstract words, leaving us with 880 words (half concrete, half abstract) per movie on average. Mean concreteness rating for the resulting set of concrete words was 3.22, compared to 1.83 for abstract words. The final mean log frequency and mean length for the final set of concrete words was 3.69 and 4.91 compared to 3.46 and 5.27 for the final set of abstract words and were not significantly different as determined by t-tests (all p's>0.45). Surprisal given preceding linguistic context (within the same time window) was extracted with a customized script making use of the predictive processing nature of GPT-2 (*Radford et al., 2019*), which has been shown to be a good model of processing difficulty (*Wilcox et al., 2021*; *Duan et al., 2020*; *Merkx and Frank, 2021*; *Schrimpf et al., 2021*). Mean surprisal of concrete words was 22.19 bits, mean surprisal of abstract words was 21.86 bits. A t-test revealed that this difference was not significant (t=1.23, p=0.218). Mean semantic diversity of concrete words was 1.92 and 1.96 of abstract words. This difference was also not significant (t=−1.20, p=0.230).

We used luminance and loudness to control for visual and acoustic properties of the movies that might vary more or less for concrete or abstract words. These 'low-level' features might be correlated with other potentially confounding auditory and visual variables. For example, luminance correlates significantly with stimulus intensity and contrast (*Johnson and Casson, 1995*) and loudness correlates with pitch (*Wengenroth et al., 2014*), prosody (*Couper-Kuhlen, 2004*), and speaking rate (*Kuhlmann et al., 2022*). Thus, luminance and loudness for each frame in the movie was measured using the 'Librosa' package for music and audio analysis in Python (*McFee et al., 2015*). We then averaged these measures across the full duration of each word. Mean luminance for concrete words was 0.72, compared to 0.65 for abstract words. These were significantly different (t(4798)=9.13 p<0.001). The mean loudness for concrete words was 0.69, compared to 0.77 for abstract words. These were also significantly different (t(4798)=9.86, p<0.001).

For the analysis looking at conceptual processing within context, we similarly wanted to check for collinear variables in the 2 s context window preceding each word, which could have confounding effects. In particular, we looked at surprisal given linguistic context, as well as the visual variables motion (optical flow), color saturation, and spatial frequency. We extracted the visual features for each frame in the 2 s context window preceding each label using the scikit-image package (*Walt et al., 2014*).

## Conceptual processing across contexts

In this analysis, we tested the prediction that when contextual information is averaged away, the neurobiological organization of conceptual processing will reflect brain systems involved in experiential and linguistic information processing, broadly in line with previous studies. Specifically, sensory and motor system engagement for concrete concepts and internal/interoceptive/affective and more

general linguistic processing system engagement for abstract concepts. All statistical analyses on the preprocessed NiFTI files were carried out in AFNI (*Cox, 1996*; *Cox and Hyde, 1997*). Individual AFNI programs used are indicated parenthetically or in italics in subsequent descriptions.

## Deconvolution analysis

We used an amplitude (also known as parametric) modulated deconvolution regression to estimate activity associated with concrete and abstract words from the preprocessed fMRI data. Specifically, we estimated four sets of amplitude-modulated impulse response functions (IRF) for (1) abstract words; (2) concrete words; (3) remaining words; and (4) other time points. Both concrete and abstract words included word onset and five modulators, two of interest, and five nuisance modulators. These were the independent ratings of concreteness and abstractness and luminance, loudness, duration, word frequency, and speaking rate (calculated as the number of phonemes divided by duration) for each word. We also estimated the IRFs in the same manner and with the same amplitude modulators for all the remaining words in the movie that were not of interest to our hypothesis. Finally, we generated IRFs (without amplitude modulators) for all time points which did not include any speech. The deconvolution model also included general linear tests for (1) abstract words under the curve; (2) concrete words under the curve; (3) contrasts between concrete and abstract words at each timepoint (for a comparison of significant timing differences see *Figure 1—figure supplement 2*).

In contrast to a standard convolution-based regression analysis, deconvolution does not assume a canonical hemodynamic response function. Instead, an IRF is estimated over a 20 s time-window from stimulus onset at 1 s steps using multiple basis functions. This produces a better understanding of shape differences between individual hemodynamic response functions and achieves higher statistical power at both individual and group-levels (*Chen et al., 2015*). Furthermore, there might be differences in timing for the processing of concrete and abstract words (*Kroll and Merves, 1986*). In particular, the 'concreteness effect' indicates that concrete words are processed faster and more accurately than abstract words (*Paivio et al., 1994*; *Jessen et al., 2000*; *Fliessbach et al., 2006*). These timing differences can be captured by our approach. We chose a 20 s time window because this should be sufficient to capture the hemodynamic response function for each word. We selected 'Csplin' over the Tent function to deconvolve the BOLD signal because this function offers more interpolation between time points, which might result in a more precise estimate of the individual response function (but is computationally more costly).

Furthermore, traditional 'main effect' analysis confounds various non-specific processes, such as acoustic processing, which co-vary with each presented word (especially during dynamic, naturalistic stimuli). In contrast, amplitude modulation allows us to isolate regions that exhibit activational fluctuations specifically in relation to the concreteness modulator beyond the 'main effect' and fluctuations in the amplitude response caused by other modulators included in our model. Including nuisance modulators can help serve as controls, mitigating potentially confounding effects - in our case the significant differences between luminance and loudness. By adjusting for these sensory attributes, we ensure that the final betas from this analysis represent the estimated BOLD response specifically associated with concreteness.

## Group-level analysis

We then used the 20 amplitude-modulated beta-coefficients from the concrete-abstract contrasts in a linear mixed effects model for group-level analysis using '*3dLME*' (*Chen et al., 2013*). The model included the factors 'contrast' with levels 'abstract' and 'concrete' and 'time' with 20 levels. The model also included a centered covariate for age of participant, and covariates for gender (two levels) and movie ID (10 levels). Besides a random intercept for participant we included a control implemented in '*3dLME*' for the potentially auto-correlative structure of residuals to make sure that we model the true effect estimates of the multiple basis function (*Hefley et al., 2017*). The final model formula was: *contrast + age + gender + movie*. We included 20 general linear tests, one for each contrast between concrete and abstract activation at each of the 20 timepoints, because we wanted to see how the amplitude of the activation associated with concreteness and abstractness changes over time. We thought that the timing and/or amplitude of the response for concrete and abstract words might vary and that this might be particularly true of the subsequent context analysis. We provide information on timing differences in the supplementary material (*Figure 3—figure supplement 7*).

## Correction for multiple comparisons

To correct for multiple comparisons in the LME, we used a multi-threshold approach rather than choosing an arbitrary p value at the individual voxel level threshold. In particular, we used a cluster simulation method to estimate the probability of noise-only clusters using the spatial autocorrelation function from the LME residuals ('*3dFWHMx*' and '*3dClustSim*'). This resulted in the cluster sizes to achieve a corrected alpha value of 0.01 at 9 different p values (i.e. 0.05, 0.02, 0.01, 0.005, 0.002, 0.001, 0.0005, 0.0002, and 0.0001). We thresholded each map at the corresponding z-value for each of these nine p-values and associated cluster sizes. We then combined the resulting maps, leaving each voxel with its original z-value. For additional protection and presentation purposes, we use a minimum cluster size 20 voxels for all results using '3dMerge'. For tables, we determined the center of mass for each of these clusters using '*3dCM.*' See *Cox et al., 2017* for a validation of a related method and *Skipper et al., 2022* for an earlier application.

## Analyses of experiential features

In order to more closely characterize the functional specificity of the spatial activation maps from the preceding LME analysis, we carried out the following two additional analyses. In both, the goal is to determine whether brain activity associated with concrete and abstract word modulation relates to separable experiential domains that roughly map onto the aforementioned sensory-motor vs internal/interoceptive/affective/linguistic distinction, respectively.

### Meta-analytic descriptions

The resulting coordinates of the center of mass of each cluster associated with modulation of concreteness and abstractness were inputed into Neurosynth (https://neurosynth.org/, *Yarkoni et al., 2011*), an online tool that includes activation maps of 14,371 neuroscientific studies (accessed April, 2023). Neurosynth automatically mines all words in titles and abstracts of these articles and performs a two-way ANOVA, testing for the presence of a non-zero association between terms reporting activation that overlaps with the input location. We scraped all terms with z scores above 3.12 (p<0.001) (excluding those related to specific brain regions and nondescript terms related to methods, tasks, or results) and repeated this procedure for each concrete and abstract cluster to determine functionally associated terms. We then tested whether any of these terms were more important for concrete or abstract words across clusters using a Kruskal-Wallis test. We did not correct for multiple comparisons, as this analysis was exploratory in nature and we did not have a prediction about how many terms we would end up with.

### Peak and valley analysis

The meta-analytic approach can only provide relatively general functional descriptions of concrete and abstract words as it is based only on high frequency terms in published titles and abstracts. To provide more precise functional specificity, we used a variant of the 'reverse correlation' method (*Hasson et al., 2004*), called the 'Peaks and Valleys Analysis' (*Hasson et al., 2008*; *Skipper et al., 2009*). For each participant, this analysis averaged the time series of voxels within clusters of modulated activity associated with concreteness and abstractness and relates this directly to features of the perceived stimulus. The approach assumes that, if a brain region encodes certain features, e.g. sensorimotor features, valence, or arousal, then activity will rise (creating peaks) in that region when the feature is present in the stimulus and fall (resulting in valleys) when it is absent.

We first extracted the averaged time series for each activation cluster for the concrete and abstract modulations across voxels using *3dMerge*. Next, we determined peaks and valleys by calculating the discrete difference 'Δ' along the time series 'x' for each value 'i' using the '*Numpy*' Python package (*Harris et al., 2020*; *Figure 7*, (1)), where $\Delta x[i]=x[i+1] - x[i]$. Given that the canonical model of the hemodynamic response function is said to peak at around 6 s after stimulus onset for stimuli of our length, we extracted the words that were mentioned at each peak and valley in a given cluster's time series with a 5- and 6 s lag (*Figure 7*, (2)). We then used the Lancaster sensorimotor norms (*Lynott et al., 2020*) and norms for valence and arousal (*Warriner et al., 2013*) to determine a 13-dimensional experience-based representation for each word (*Figure 7*, (3)), which included the dimensions: 'Auditory,' 'Gustatory,' 'Haptic,' 'Interoception,' 'Visual,' 'Hand_Arm,' 'Foot_Leg,' 'Torso,' 'Mouth,' 'Head,' 'Olfactory,' 'Valence,' and 'Arousal.'

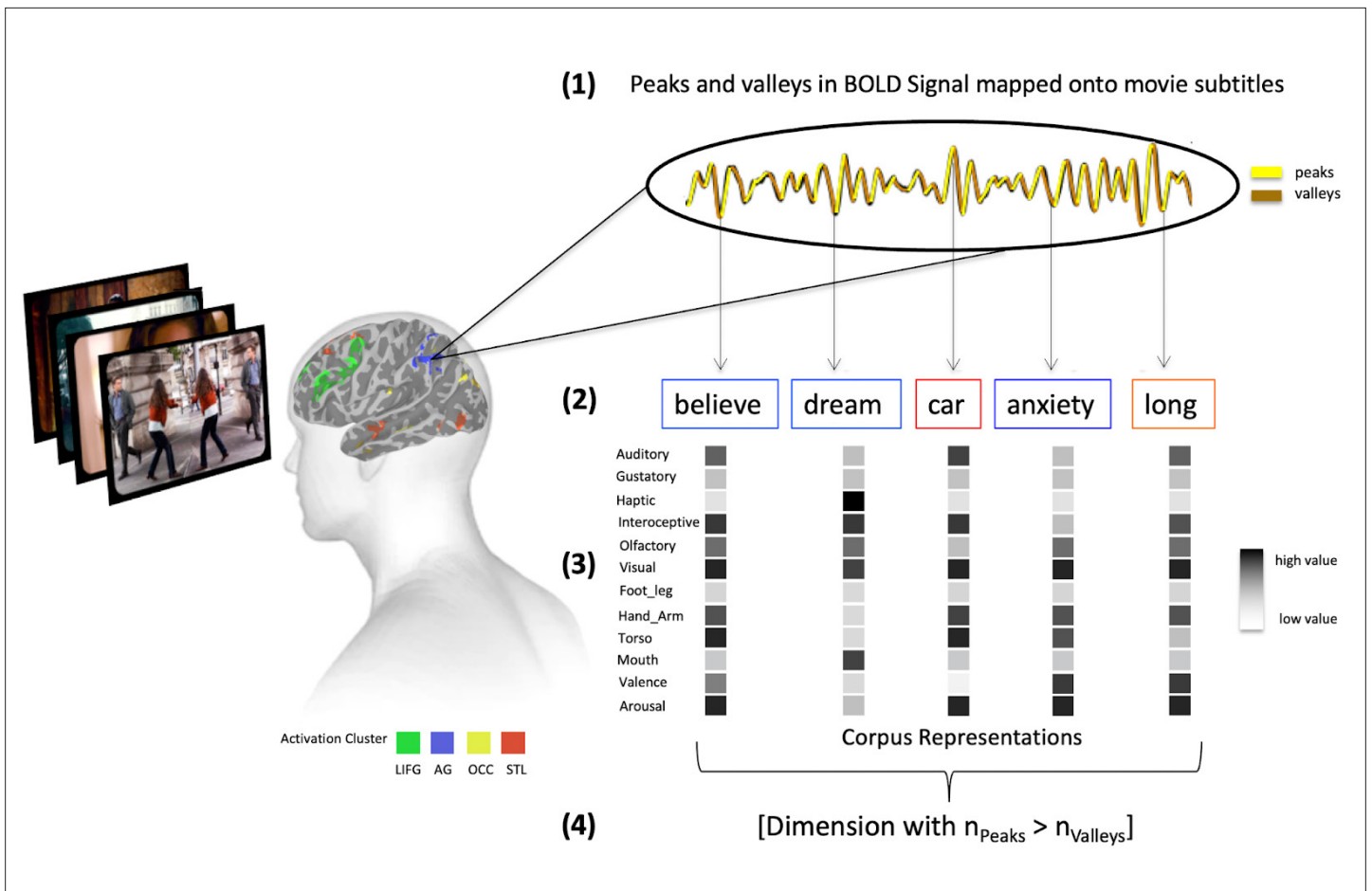

**Figure 7.** Overview of the peak and valley analysis method. First, we average the fMRI time series for each participant, for each abstract, concrete, and overlap cluster of activity from *Figure 1*. Then we label peaks and valleys in these (1) and map them onto word on- and off-set times (2). Finally, we estimate sensorimotor as well as valence and arousal representations for each abstract (blue frame) and concrete word (red frame) (3) and determine which dimensions are associated with significantly more peaks than valleys across participants in each cluster using a Kruskal-Wallis test (4).

Specifically, for concrete clusters, we expected significantly more sensory-motor features (i.e. 'Foot_Leg,' 'Hand_Arm,' 'Haptic,' 'Visual,' and 'Torso' in the corpora) to be associated with peaks rather than valleys in the time series compared to abstract clusters. Conversely, we expected significantly more experiential features related to internal/interoceptive/affective processing (i.e. 'Interoception,' 'Valence,' and 'Arousal' in the corpora) to be associated with peaks compared to valleys for abstract relative to concrete clusters. It was not clear to us whether the dimensions ('Auditory,' 'Head,' and 'Gustatory') were more related to internal/interoceptive/affective or sensory-motor processing. Therefore, we made no predictions for those.

For each of these dimensions, we created two categorical arrays, one for peaks and one for valleys, noting down 0 if the word mentioned at a peak or valley was not highly rated on the dimension and 1 if it was rated highly. This was defined as a deviation of at least one standard deviation from the mean. Given the distributional nature of this data, we then conducted a Kruskal-Wallis test between these arrays to determine whether a given experiential dimension occurred significantly more with peaks than valleys in the averaged time series of a cluster (*Figure 7*, (4)). We repeated this procedure for a 4 s, 5 s, and a 6 s time series lag and conducted a cosine-similarity test between each result using the '*Sklearn*' package in Python *Pedregosa et al., 2011* in order to determine if they were significantly different. The results for the 5 s and 6 s lag converge but not for 4 s. This was expected, because the delay of the HRF is somewhere between 5 and 6 s. In the main results, we randomly decided between presenting the 5 s and 6 s lag. The 5 s lag is now displayed in *Figure 3—figure supplement 1*, the 4 s lag in *Figure 3—figure supplement 2*.

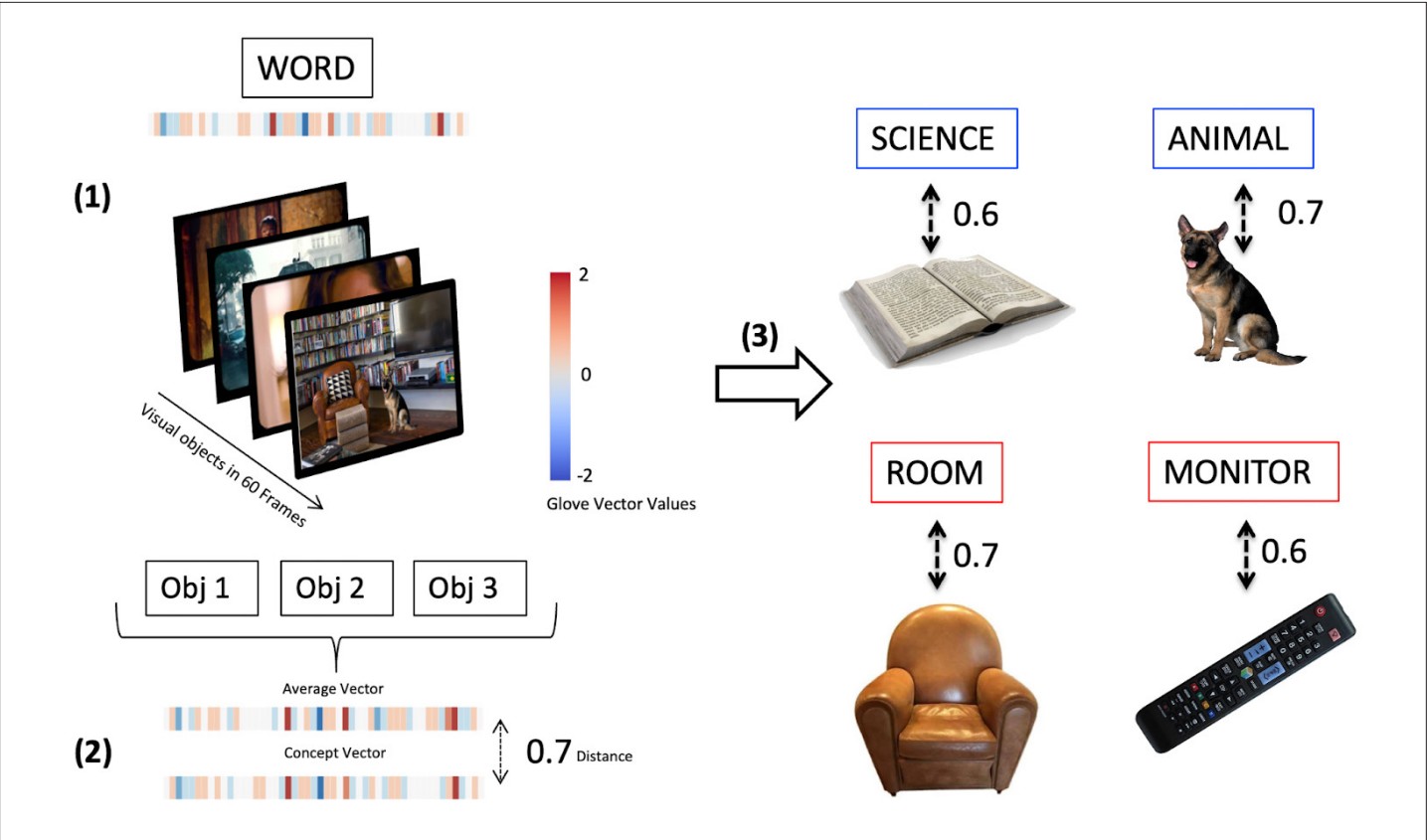

**Figure 8.** Method for estimating contextual situatedness for each concrete and abstract word to model context-dependent modulation of conceptual encoding. We use visual recognition models for automatically extracting labels that were visually present in the scene (60 frames, ~2 s) before a given word was mentioned in the movie (1). We then correlate an average GloVe Vector embedding of all these labels with a GloVe Vector embedding of that word to estimate how closely related the labels of objects in the scene are to the word (2). Displayed are four randomly extracted measures of situated abstract (blue frame) and concrete (red frame) words (3) together with the objects that were visually present in the scene.

## Conceptual processing in context

The previous analyses tested whether, when activation is considered across contexts, concrete and abstract words are processed in separable brain regions encoding different types of experiential information. Here, we test if these response profiles dynamically change depending on the objects present in the viewing context. We predict that when abstract concepts are situated in highly related contexts (e.g. the word 'love' is processed while watching two people kissing), they engage neurobiological regions that are usually involved in processing concrete concepts and are related to processing of external visual information. Conversely, when concrete concepts are displaced from the visual context (e.g. processing the word 'apple' while watching the interior of a house), we predict them to engage more abstract-like regions in the brain that are related to processing of internal/interoceptive/affective information. Note that we chose to do the analysis in two stages (first across context, then within context) because only a subset of the 440 words used in our analysis across context for concrete and abstract words are related to the objects present in the scene in a way that situates them in visual context (see below). We wanted to have as much power as possible for the first deconvolution/LME to look beyond the effects of context.

### Estimating contextual situatedness

To test our predictions, we estimated a measure of contextual situatedness for each concrete and abstract word included in the first analysis. To that end, we utilized two pre-trained visual recognition models, Faster R-CNN (*He et al., 2015*) and OmniSource (*Duan et al., 2020*), to extract object features using computer vision toolboxes (*Mkrtychian et al., 2019*), respectively. For each prediction frame (about every four frames, i.e. 4*0.04=0.16 s), the object recognition model generated a list of

detected object labels and kept those that had a prediction confidence of 90% or greater (**Figure 8**, 1). Then, we excluded all objects that were recognized at least three standard deviations more often by the model compared to the mean recognition rate of objects (which was 682 appearances), because they would bias our measure of situatedness. These labels were 'person' (17,856 appearances per movie on average), 'chair' (9718 appearances per movie on average), and 'tie' (8123 appearances per movie on average). After exclusion of these labels, the final object features were represented as the average of the vectorized object labels using GloVe (**Pennington et al., 2014**; **Figure 8**, 2), which represents the meaning of each label via global co-occurrence statistics.

We then estimated a representation of a 2 s (or 60 frames) context window, which should capture the immediate visual context leading up to each concrete and abstract word. We extracted all the labels of objects visually present in each frame within that window. Finally, we calculated the cosine similarity between the vector representation of each word and its context average, using the '*Sklearn*' package in Python (**Pedregosa et al., 2011**), to estimate contextual situatedness for each concrete and abstract word (a value $c$ between 0 and 1), **Figure 8**, (3). After separating into situated ($c>0.6$) and displaced words ($c<0.4$), we were left with (on average for each movie) 164 abstract situated words, 201 abstract displaced words, 215 concrete situated words, and 172 concrete displaced words. Given that, as concerns observations, the abstract situated condition was the limiting condition, we randomly selected words from the abstract displaced, concrete situated, and concrete displaced conditions to have an equal number of 164 words in each condition (on average per movie).

Though we use visual nuisance regressors, we note that there may be additional confounding visual information when estimating contextual situatedness: high situatedness may correlate positively with the number of objects present and, therefore, 'naturally' engage visual processing more for abstract situated concepts. To alleviate this concern, we counted the number of objects in the abstract situated (8315 objects across movies) and abstract displaced (7443 across movies) conditions. The difference between the two (872) is not statistically significant ($H(2)=4.1$, $p<0.09$).

## Deconvolution analysis

The deconvolution analysis was as described previously except that the sets of concrete and abstract words were broken into four equal subsets of regressors, i.e., situated concrete, displaced concrete, situated abstract, and displaced abstract words and modulators with 164 words per each set. The four contrasts included were: (1) abstract situated vs abstract displaced, (2) concrete situated vs concrete displaced, (3) abstract situated vs concrete situated, and (4) abstract displaced vs concrete displaced. Mean concreteness rating for the resulting sets of concrete words were 3.39 for displaced words and 3.35 for situated words, compared to 1.84 for abstract situated words and 1.71 for abstract displaced words. The mean log frequency and mean length for concrete words was 4.91 and 4.88 for situated words and 5.33 and 4.91 for displaced words, compared to 5.11 and 5.08 for abstract situated words and 5.20 and 5.27 for abstract displaced words. Mean surprisal ratings for concrete situated words were 21.98 bits, 22.02 bits for the displaced concrete words, 22.10 for the situated abstract words and 22.25 for the abstract displaced words. Mean semantic diversity ratings were 1.88 for the concrete situated words, 2.19 for the concrete displaced words, 2.03 for the abstract situated words, and 1.95 for the abstract displaced words. As concerns visual variables, mean optical flow was 0.85, mean color saturation was 0.33, and mean spatial frequency was 0.02 for the 2 s context windows before abstract situated words. Mean optical flow was 0.81, mean color saturation was 0.35 and mean spatial frequency was 0.04 for the 2 s context windows before abstract displaced words. Mean optical flow was 0.76, mean color saturation was 0.41 and mean spatial frequency was 0.10 for the 2 s context windows before concrete situated words. Mean optical flow was 0.86, mean color saturation was 0.33, and mean spatial frequency was 0.05 for the 2 s context windows before concrete displaced words. Pairwise T-tests between all of the mentioned measures for all groups revealed no significant differences.

## Group-level analysis

A linear mixed effects model for group-level analysis was conducted on the 20 amplitude-modulated betas from each condition (concrete situated, abstract situated, concrete displaced, situated displaced) using '*3dLME*' (**Chen et al., 2013**). The model included factors 'word_type' (concrete and abstract), 'context' (displaced and situated), and 'time' (20 levels) and all possible interactions between these

factors. We again included covariates for age, gender, and movie, a random intercept for participant, and a control of the auto-correlative structure of residuals. The final model formula was: *word_type \* context \* time + age + gender + movie*. We had no prediction on whether the timing and/or amplitude of the response for concrete and abstract words would vary in the present analysis. Therefore, we included four general linear tests, one for each contrast across time: one for the abstract situated vs. abstract displaced contrast, one for the concrete situated vs. concrete displaced contrast, one for the abstract situated vs abstract displaced contrast, and one for the concrete situated vs concrete displaced contrast.

## Code

We implemented our data analysis in Bash, Python, and R. Our code will be provided as online supplemental material upon publication and hosted openly on a dedicated Github repository under: (https://github.com/ViktorKewenig/Naturalistic_Encoding_Concepts, copy archived at *ViktorKewenig, 2024*).

## Acknowledgements

VK would like to thank Bangjie Wang for his help in using image recognition software, and Dr. Sarah Aliko for help with neuroimaging analysis. This work was supported in part by the European Research Council Advanced Grant (ECOLANG, 743035); Royal Society Wolfson Research Merit Award (WRM\R3\170016) to GV; and Leverhulme award DS-2017–026 to VK and GV.

## Additional information

### Funding

| Funder | Grant reference number | Author |
| --- | --- | --- |
| European Research Council | ECOLANG,743035 | Gabriella Vigliocco |
| Royal Society | WRM\R3\170016 | Gabriella Vigliocco |
| Leverhulme Trust | DS-2017-026 | Viktor Nikolaus Kewenig |

The funders had no role in study design, data collection and interpretation, or the decision to submit the work for publication.

### Author contributions

Viktor Nikolaus Kewenig, Conceptualization, Software, Funding acquisition, Visualization, Methodology, Writing - original draft, Writing - review and editing; Gabriella Vigliocco, Jeremy I Skipper, Conceptualization, Supervision, Writing - review and editing

### Author ORCIDs

Viktor Nikolaus Kewenig ![ORCID] https://orcid.org/0009-0009-5912-0676
Jeremy I Skipper ![ORCID] https://orcid.org/0000-0002-5503-764X

### Ethics

The study was approved by the ethics committee of University College London and participants provided written informed consent to take part in the study and share their anonymised data. Ethics Committee (Reference Number 143/003).

Reviewer #1 (Public review): https://doi.org/10.7554/eLife.91522.3.sa1
Reviewer #2 (Public review): https://doi.org/10.7554/eLife.91522.3.sa2
Reviewer #3 (Public review): https://doi.org/10.7554/eLife.91522.3.sa3
Author response https://doi.org/10.7554/eLife.91522.3.sa4

# Additional files

## Supplementary files
• MDAR checklist

## Data availability
All code and data are publicly available.

The following previously published dataset was used:

| Author(s) | Year | Dataset title | Dataset URL | Database and Identifier |
|-----------|------|---------------|-------------|-------------------------|
| Aliko S, Huang J, Meliss S, Skipper JI | 2020 | A naturalistic neuroimaging database | https://openneuro.org/datasets/ds002837/versions/2.0.0 | OpenNeuro, 10.18112/openneuro.ds002837.v2.0.0 |

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
