## [Editor Report · eLife assessment]

Kewenig et al. present a timely and **valuable** study that extends prior research investigating the neural basis of abstract and concrete concepts by examining how these concepts are processed in a naturalistic stimulus: during movie watching. The authors provide **convincing** evidence that the varying strength of the relationship between a word and a particular visual scene is associated with a change in the similarity between the brain regions active for concrete and abstract words. This work makes a contribution that will be of general interest within any field that faces the inherent challenge of quantifying context in a multimodal stimulus.

---

## [Referee Report · Reviewer #1 (Public review)]

Summary:

In this study, the authors investigate a very interesting but often overlooked aspect of abstract vs. concrete processing in language. Specifically, they study if the differences in processing of abstract vs. concrete concepts in the brain is static or dependent on the (visual) context in which the words occur. This study takes a two-step approach to investigate how context might affect the perception of concepts. First, the authors analyze if concrete concepts, expectedly, activate more sensory systems while abstract concepts activate higher-order processing regions. Second, they measure the contextual situatedness vs. displacement of each word with respect to the visual scenes in which it is spoken and then evaluate if this contextual measure correlates with more activation in the sensory vs. higher-order regions respectively.

Strengths:

This study raises a pertinent and understudied question in language neuroscience. It also combines both computational and meta-analytic approaches.

---

## [Referee Report · Reviewer #2 (Public review)]

Summary:

This study tests a plausible and intriguing hypothesis that one cause of the differences in the neural underpinnings of concrete and abstract words is differences in their grounding in the current sensory context. The authors reasoned that, in this case, an abstract word presented with a relevant visual scene would be processed in a more similar way to a concrete word. Typically, abstract and concrete words are tested in isolation. In contrast, this study takes advantage of naturalistic movie stimuli to assess the neural effects of concreteness in both abstract and concrete words (the speech within the film), when the visual context is more or less tied to the word meaning (measured as the similarity between the word co-occurrence-based vector for the spoken word and the average of this vector across all present objects). This novel approach allows a test of the dynamic nature of abstract and concrete word processing, and as such provides a useful perspective accounting for differences in processing these word types.

The measure of contextual situatedness (how related a spoken word is to the average of the visually presented objects in a scene) is an interesting approach allowing parametric variation within naturalistic stimuli, which is a potential strength of the study. Additionally, the authors use an interesting peak and valley method and provide a rationale for this approach. This provided additional temporal information on the processing of spoken concrete and abstract words.

The authors predicted that sensory areas would be more active for concrete words, affective areas for abstract and language areas would be involved in both. The use of reverse inference to interpret areas such as the inferior frontal gyrus post hoc, as sensory, affective or language related deserves some caution. It is also important to remember that the different areas identified for each comparison do not necessarily have the same roles. As the number of clusters may therefore be a misleading way to assess the relationship of these areas with the sensory terms, the relationship between each area and the different sensory terms is provided in the supplemental to allow more nuanced interpretation. The study could benefit from being better situated in the prior literature on context and concrete vs abstract regional differences. Overall, the authors successfully demonstrate the context-dependent nature of abstract and concrete word processing.

---

## [Referee Report · Reviewer #3 (Public review)]

Summary:

The primary aim of this manuscript was to investigate how context, defined from visual object information in multimodal movies, impacts the neural representation of concrete and abstract conceptual knowledge. The authors first conduct a series of analyses to identify context independent regional response to concrete and abstract concepts in order to compare these results with the networks observed in prior research using non-naturalistic paradigms. The authors then conduct analyses to investigate whether regional response to abstract and concrete concepts changes when the concepts are either contextually situated or displaced. A concept is considered displaced if the visual information immediately preceding the word is weakly associated with the word whereas a concept is situated if the association is high. The results suggest that, when ignoring context, abstract and concrete concepts engage different brain regions with overlap in core language areas. When context is accounted for, however, similar brain regions are activated for processing concrete and situated abstract concepts and for processing abstract and displaced concrete concepts. The authors suggest that contextual information dynamically changes the brain regions that support the representation of abstract and concrete conceptual knowledge.

Strengths:

There is significant interest in understanding both the acquisition and neural representation of abstract and concrete concepts, and most of the work in this area has used highly constrained, decontextualized experimental stimuli and paradigms to do so. This manuscript addresses this limitation by using multimodal narratives which allows for an investigation of how context-sensitive the regional response to abstract and concrete concepts is. The authors characterize the regional response in a comprehensive way.

Weaknesses:

The edits made to the manuscript in response to the reviewer comments have clarified and strengthened the methodological concerns flagged by all reviewers, giving me greater confidence that the authors are capturing what they aimed to and are making appropriate inferences given the results.

---

## [Author Response]

The following is the authors’ response to the original reviews.

We thank the reviewers and the editorial team for a thoughtful and constructive assessment. We appreciate all comments, and we try our best to respond appropriately to every reviewer’s queries below. It appears to us that one main worry was regarding appropriate modelling of the complex and rich structure of confounding variables in our movie task.

One recent approach fits large feature vectors that include confounding variables along the variable(s) of interest to the activity of each voxel in the brain to disentangle the contributions of each variable to the total recorded brain response. While these encoding models have yielded some interesting results, they have two major drawbacks which makes using them unfeasible for our purposes (as we explain in more detail below): first, by fitting large vectors to individual voxels, they tend to over-estimate effect size; second, they are very ineffective at unveiling group-level effects due to high variability between subjects. Another approach able to deal with at least the second of these worries is “inter-subject-correlation”. In this technique brain responses are recorded from multiple subjects while they are presented with natural stimuli. For each brain area, response time courses from different subjects are correlated to determine whether the responses are similar across subjects. Our “peak and valley” analysis is a special case of this analysis technique, as we explain in the manuscript and below.

For estimating individual-level brain-activation, we opted for an approach that adapts a classical method of analysing brain data – convolution - to naturalistic settings. Amplitude modulated deconvolution extends classical brain analysis tools in several ways to handle naturalistic data:

(1) The method does not assume a fixed hemodynamic response function (HRF). Instead, it estimates the HRF over a specified time window from the data, allowing it to vary in amplitude based on the stimulus. This flexibility is crucial for naturalistic stimuli, where the timing and nature of brain responses can vary widely.

(2) The method only models the modulation of the amplitude of the HRF above its average with respect to the intensity or characteristics of the stimulus.

(3) By allowing variation in the response amplitude, non-linear relationships between the stimulus and brain-response can be captured.

It is true that amplitude modulated deconvolution does not come without its flaws – for example including more than a few nuisance regressors becomes computationally very costly. Getting to grips with naturalistic data (especially with fMRI recordings) continuous to be an active area of research and presents a new and exciting challenge. We hope that we can convince reviewers and editors with this response and the additional analyses and controls performed, that the evidence presented for the visual context dependent recruitment of brain areas for abstract and concrete conceptual processing is not incomplete.

Overview of Additional Analyses and Controls Performed by the Authors:

(1) Individual-Level Peaks and Valleys Analysis (Supplementary Material, Figures S3, S4, and S5)

(2) Test of non-linear correlations of BOLD responses related to features used in the Peak and Valley Analysis (Supplementary Material, Figures S6, S7)

(3) Comparison of Psycholinguistic Variables Surprisal and Semantic Diversity between groups of words analysed (no significant differences found)

(4) Comparison of Visual Variables Optical Flow, Colour Saturation, and Spatial Frequency for 2s Context Window between groups of words analysed (no significant differences found)

These controls are in addition to the five low-level nuisance regressors included in our model, which are luminance, loudness, duration, word frequency, and speaking rate (calculated as the number of phonemes divided by duration) associated with each analysed word.

**Public Reviews:**

**Reviewer #1 (Public Review):**
Peaks and Valleys Analysis:(1) Doesn't this method assume that the features used to describe each word, like valence or arousal, will be linearly different for the peaks and valleys? What about non-linear interactions between the features and how they might modulate the response?

Within-subject variability in BOLD response delays is typically about 1 second at most (Neumann et al., 2003). As individual words are presented briefly (a few hundred Ms at most) and the BOLD response to these stimuli falls within that window (1s/TR), any nonlinear interactions between word features and a participant’s BOLD response within that window are unlikely to significantly affect the detection of peaks and valleys.

To quantitatively address the concern that non-linear modulations could manifest outside of that window, we include a new analysis in Figure S6, which compares the average BOLD responses of each participant in each cluster and each combination of features, showing that only a very few of all possible comparisons differ significantly from each other (~ 5000 combinations of features were significantly different from each other given an overall number of ~130.000 comparisons between BOLD responses to features, which amounts to 3.85%), suggesting that there are no relevant non-linear interactions between features. For a full list of the most non-linearly interacting features see Figure S7.

(2) Doesn't it also assume that the response to a word is infinitesimal and not spread across time? How does the chosen time window of analysis interact with the HRF? From the main figures and Figures S2-S3 there seem to be differences based on the timelag.

The Peak and Valley (P&V) method does not assume that the response to a word is infinitesimal or confined to an instantaneous moment. The units of analysis (words) fall within one TR, as they are at most hundreds of Ms long – for this reason, we are looking at one TR only. The response of each voxel at that TR will be influenced by the word of interest, as well as all other words that have been uttered within the 1s TR, and the multimodal features of the video stimulus that fall within that timeframe. So, in our P&V, we are not looking for an instantaneous response but rather changes in the BOLD signal that correspond to the presence of linguistic features within the stimuli.

The chosen time window of analysis interacts with the human response function (HRF) in the following way: the HRF unfolds over several seconds, typically peaking around 5-6 seconds after stimulus onset and returning to baseline within 20-30 seconds (Handwerker et al., 2004).

Our P&V is designed to match these dynamics of fMRI data with the timing of word stimuli. We apply different lags (4s, 5s, and 6s) to account for the delayed nature of the HRF, ensuring that we capture the brain's response to the stimuli as it unfolds over time, rather than assuming an immediate or infinitesimal effect. We find that the P&V yields our expected results for a 5s and a 6s lag, but not a 4s lag. This is in line with literature suggesting that the HRF for a given stimulus peaks around 5-6s after stimulus onset (Handwerker et al., 2004). As we are looking at very short stimuli (a few hundred ms) it makes sense that the distribution of features would significantly change with different lags. The fact that we find converging results for both a 5s and 6s lag, suggests that the delay is somewhere between 5s and 6s. There is no way of testing this hypothesis with the resolution of our brain data, however (1 TR).

(3) Were the group-averaged responses used for this analysis?

Yes, the response for each cluster was averaged across participants. We now report a participant-level overview of the Peak and Valley analysis (lagged at 5s) with similar results as the main analysis in the supplementary material see Figures S3, S4, and S5.

(4) Why don't the other terms identified in Figure 5 show any correspondence to the expected categories? What does this mean? Can the authors also situate their results with respect to prior findings as well as visualize how stable these results are at the individual voxel or participant level? It would also be useful to visualize example time courses that demonstrate the peaks and valleys.

The terms identified in figure 5 are sensorimotor and affective features from the combined Lancaster and Brysbaert norms. As for the main P&V analysis, we only recorded a cluster as processing a given feature (or term) when there were significantly more instances of words highly rated in that dimension occurring at peaks rather than valleys in the HRF. For some features/terms, there were never significantly more words highly rated on that dimension occurring at peaks compared to valleys, which is why some terms identified in figure 5 do not show any significant clusters. We have now also clarified this in the figure caption.

We situate the method in previous literature in lines 289 – 296. In essence, it is a variant of the well-known method called “reverse correlation” first detailed in Hasson et al., 2004 (reference from the manuscript) and later adapter to a peak and valley analysis in Skipper et al., 2009 (reference from the manuscript).

We now present a more fine-grained characterisation of each cluster on an individual participant level in the supplementary material. We doubt that it would be useful to present an actual example time-course as it would only represent a fraction of over one hundred thousand analysed time-series. We do already present an exemplary time-course to demonstrate the method in Figure 1.

Estimating contextual situatedness:(1) Doesn't this limit the analyses to "visual" contexts only? And more so, frequently recognized visual objects?

Yes, it was the point of this analysis to focus on visual context only, and it may be true that conducting the analysis in this way results in limiting it to objects that are frequently recognized by visual convolutional neural networks. However, the state-of-the-art strength of visual CNNs in recognising many different types of objects has been attested in several ways (He et al., 2015). Therefore, it is unlikely that the use of CNNs would bias the analysis towards any specific “frequently recognised” objects.

(2) The measure of situatedness is the cosine similarity of GloVe vectors that depend on word co-occurrence while the vectors themselves represent objects isolated by the visual recognition models. Expectedly, "science" and the label "book" or "animal" and the label "dog" will be close. But can the authors provide examples of context displacement? I wonder if this just picks up on instances where the identified object in the scene is unrelated to the word. How do the authors ensure that it is a displacement of context as opposed to the two words just being unrelated? This also has a consequence on deciding the temporal cutoff for consideration (2 seconds).

The cosine similarity is between the GloVe vectors of the *word* (that is situated or displaced) and the *words* referring to the objects identified by the visual recognition model. Therefore, the correlation is between more than just two vectors and both correlated representations depend on co-occurrence. The cosine similarity value reported is not from a comparison between GloVe vectors and vectors that are (visual) representations of objects from the visual recognition model.

A *word* is displaced if *all* the identified *object-words* in the defined context window (2s before word-onset) are unrelated to the _word (_see lines 105-110 (pg. 5); lines 371-380 pg. 1516 and Figure 2 caption). Thus, a word is considered to be displaced if *all* identified objects (not just two as claimed by the reviewer) in the scene are unrelated to the word. Given a context of 60 frames and an average of 5 identified objects per frame (i.e. an average candidate set of 300 objects that could be related) per word, the bar for “displacement” is set high. We provide some further considerations justifying the context window below in our responses to reviewers 2 and 3.

(3) While the introduction motivated the problem of context situatedness purely linguistically, the actual methods look at the relationship between recognized objects in the visual scene and the words. Can word surprisal or another language-based metric be used in place of the visual labeling? Also, it is not clear how the process identified in (2) above would come up with a high situatedness score for abstract concepts like "truth".

We disagree with the reviewer that the introduction motivated the problem of context situatedness purely linguistically, as we explicitly consider *visual context* in the abstract as well as the introduction. Examples in text include lines 71-74 and lines 105-115. This is also reflected in the cited studies that use visual context, including Kalenine et al., 2014; Hoffmann et al., 2013; Yee & Thompson-Schill, 2016; Hsu et al., 2011. However, we appreciate the importance of being very clear about this point, so we added various mentions of this fact at the beginning of the introduction to avoid confusion.

We know that prior linguistic context (e.g. measured by surprisal) does affect processing. The point of the analysis was to use a non-language-based metric of visual context to understand how this affects conceptual representation in naturalist settings. Therefore, it is not clear to us why replacing this with a language-based metric such as surprisal would be an adequate substitution. However, the reviewer is correct that we did not control for the influence of prior context. We obtained surprisal values for each of our words but could not find any significant differences between conditions and therefore did not include this factor in the analyses conducted. For considerations of differences in surprisal between each of the analysed sets of words, see the supplementary material.

The method would yield a high score of contextual situatedness for abstract concepts if there were objects in the scene whose GloVe embeddings have a close cosine distance to the GloVe embedding of that abstract word (e.g., “truth” and “book”). We believe this comment from the reviewer is rooted in a misconception of our method. They seem to think we compared GloVe vectors for the spoken word with vectors from a visual recognition model directly (in which case it is true that there would be a concern about how an abstract concept like “truth” could have a high situatedness). Apart from the fact that there would be concerns about the comparability of vectors derived from GloVe and a visual recognition model more generally, this present concern is unwarranted in our case, as we are comparing GloVe embeddings.

(4) It is a bit hard to see the overlapping regions in Figures 6A-C. Would it be possible to show pairs instead of triples? Like "abstract across context" vs. "abstract displaced"? Without that, and given (2) above, the results are not yet clear. Moreover, what happens in the "overlapping" regions of Figure 3?

To make this clearer, we introduced the contrasts (abstract situated vs displaced and concrete situated vs displaced) that were previously in the supplementary materials in the main text (now Figure 6, this was also requested by reviewer 2). We now show the overlap between the abstract situated (from the contrast in Figure 6) with concrete across context and the overlap between concrete displaced (from the contrast in Figure 6) with abstract across context separately in Figure 7.

The overlapping regions of Figure 3 indicate that both concrete and abstract concepts are processed in these regions (though at different time-points). We explain why this is a result of our deconvolution analysis on page 23:

“Finally, there was overlap in activity between modulation of both concreteness and abstractness (Figure 3, yellow). The overlap activity is due to the fact that we performed general linear tests for the abstract/concrete contrast at each of the 20 timepoints in our group analysis. Consequently, overlap means that activation in these regions is modulated by both concrete and abstract word processing but at different time-scales. In particular, we find that activity modulation associated with abstractness is generally processed over a longer time-frame. In the frontal, parietal, and temporal lobes, this was primarily in the left IFG, AG, and STG, respectively. In the occipital lobe, processing overlapped bilaterally around the calcarine sulcus.”

Miscellaneous comments:(1) In Figure 3, it is surprising that the "concrete-only" regions dominate the angular gyrus and we see an overrepresentation of this category over "abstract-only". Can the authors place their findings in the context of other studies?

The Angular Gyrus (AG) is hypothesised to be a general semantic hub; therefore it is not surprising that it should be active for general conceptual processing (and there is some overlap activation in posterior regions). We now situate our results in a wider range of previous findings in the results section under “Conceptual Processing Across Context”.

“Consistent with previous studies, we predicted that across naturalistic contexts, concrete and abstract concepts are processed in a separable set of brain regions. To test this, we contrasted concrete and abstract modulators at each time point of the IRF (Figure 3). This showed that concrete produced more modulation than abstract processing in parts of the frontal lobes, including the right posterior inferior frontal gyrus (IFG) and the precentral sulcus (Figure 3, red). Known for its role in language processing and semantic retrieval, the IFG has been hypothesised to be involved in the processing of action-related words and sentences, supporting both semantic decision tasks and the retrieval of lexical semantic information (Bookheimer, 2002; Hagoort, 2005). The precentral sulcus is similarly linked to the processing of action verbs and motor-related words (Pulvermüller, 2005). In the temporal lobes, greater modulation occurred in the bilateral transverse temporal gyrus and sulcus, planum polare and temporale. These areas, including primary and secondary auditory cortices, are crucial for phonological and auditory processing, with implications for the processing of sound-related words and environmental sounds (Binder et al., 2000). The superior temporal gyrus (STG) and sulcus (STS) also showed greater modulation for concrete words and these are said to be central to auditory processing and the integration of phonological, syntactic, and semantic information, with a particular role in processing meaningful speech and narratives (Hickok & Poeppel, 2007). In the parietal and occipital lobes, more concrete modulated activity was found bilaterally in the precuneus, which has been associated with visuospatial imagery, episodic memory retrieval, and self-processing operations and has been said to contribute to the visualisation aspects of concrete concepts (Cavanna & Trimble, 2006). More activation was also found in large swaths of the occipital cortices (running into the inferior temporal lobe), and the ventral visual stream. These regions are integral to visual processing, with the ventral stream (including areas like the fusiform gyrus) particularly involved in object recognition and categorization, linking directly to the visual representation of concrete concepts (Martin, 2007). Finally, subcortically, the dorsal and posterior medial cerebellum were more active bilaterally for concrete modulation. Traditionally associated with motor function, some studies also implicate the cerebellum in cognitive and linguistic processing, including the modulation of language and semantic processing through its connections with cerebral cortical areas (Stoodley & Schmahmann, 2009).

Conversely, activation for abstract was greater than concrete words in the following regions (Figure 3, blue): In the frontal lobes, this included right anterior cingulate gyrus, lateral and medial aspects of the superior frontal gyrus. Being involved in cognitive control, decision-making, and emotional processing, these areas may contribute to abstract conceptualization by integrating affective and cognitive components (Shenhav et al., 2013). More left frontal activity was found in both lateral and medial prefrontal cortices, and in the orbital gyrus, regions which are key to social cognition, valuation, and decision-making, all domains rich in abstract concepts (Amodio & Frith, 2006). In the parietal lobes, bilateral activity was greater in the angular gyri (AG) and inferior parietal lobules, including the postcentral gyrus. Central to the default mode network, these regions are implicated in a wide range of complex cognitive functions, including semantic processing, abstract thinking, and integrating sensory information with autobiographical memory (Seghier, 2013). In the temporal lobes, activity was restricted to the STS bilaterally, which plays a critical role in the perception of intentionality and social interactions, essential for understanding abstract social concepts (Frith & Frith, 2003). Subcortically, activity was greater, bilaterally, in the anterior thalamus, nucleus accumbens, and left amygdala for abstract modulation. These areas are involved in motivation, reward processing, and the integration of emotional information with memory, relevant for abstract concepts related to emotions and social relations (Haber & Knutson, 2010, Phelps & LeDoux, 2005).

Finally, there was overlap in activity between modulation of both concreteness and abstractness (Figure 3, yellow). The overlap activity is due to the fact that we performed general linear tests for the abstract/concrete contrast at each of the 20 timepoints in our group analysis. Consequently, overlap means that activation in these regions is modulated by both concrete and abstract word processing but at different time-scales. In particular, we find that activity modulation associated with abstractness is generally processed over a longer time-frame (for a comparison of significant timing differences see figure S9). In the frontal, parietal, and temporal lobes, this was primarily in the left IFG, AG, and STG, respectively. Left IFG is prominently involved in semantic processing, particularly in tasks requiring semantic selection and retrieval and has been shown to play a critical role in accessing semantic memory and resolving semantic ambiguities, processes that are inherently time-consuming and reflective of the extended processing time for abstract concepts (Thompson-Schill et al., 1997; Wagner et al., 2001; Hofman et al., 2015). The STG, particularly its posterior portion, is critical for the comprehension of complex linguistic structures, including narrative and discourse processing. The processing of abstract concepts often necessitates the integration of contextual cues and inferential processing, tasks that engage the STG and may extend the temporal dynamics of semantic processing (Ferstl et al., 2008; Vandenberghe et al., 2002). In the occipital lobe, processing overlapped bilaterally around the calcarine sulcus, which is associated with primary visual processing (Kanwisher et al., 1997; Kosslyn et al., 2001).”

The finding that concrete concepts activate more brain voxels compared to abstract concepts is generally aligned with existing research, which often reports more extensive brain activation for concrete versus abstract words. This is primarily due to the richer sensory and perceptual associations tied to concrete concepts - see for example Binder et al., 2005 (figure 2 in the paper). Similarly, a recent meta-analysis by Bucur & Pagano (2021) consistently found wider activation networks for the “concrete > abstract” contrast compared to the “abstract > concrete contrast”.

(2) The following line (Pg 21) regarding the necessary differences in time for the two categories was not clear. How does this fall out from the analysis method?- Both categories overlap **(though necessarily at different time points)** in regions typically associated with word processing -

This is answered in our response above to point (4) in the reviewer’s comments. We now also provide more information on the temporal differences in the supplementary material (Figure S9).

**Reviewer #2 (Public Review):**
The critical contrasts needed to test the key hypothesis are not presented or not presented in full within the core text. To test whether abstract processing changes when in a situated context, the situated abstract condition would first need to be compared with the displaced abstract condition as in Supplementary Figure 6. Then to test whether this change makes the result closer to the processing of concrete words, this result should be compared to the concrete result. The correlations shown in Figure 6 in the main text are not focused on the differences in activity between the situated and displaced words or comparing the correlation of these two conditions with the other (concrete/abstract) condition. As such they cannot provide conclusive evidence as to whether the context is changing the processing of concrete/abstract words to be closer to the other condition. Additionally, it should be considered whether any effects reflect the current visual processing only or more general sensory processing.

The reviewer identifies the critical contrast as follows:

“The situated abstract condition would first need to be contrasted with the displaced abstract condition. Then, these results should be compared to the concrete result.”

We can confirm that this is indeed what had been done and we believe the reviewer’s confusion stems from a lack of clarity on our behalf. We have now made various clarifications on this point in the manuscript, and we changed the figures to make clear that our results are indeed based on the contrasts identified by this reviewer as the essential ones.

Figure 6 in the main text now reflects the contrast between situated and displaced abstract and concrete conditions (as requested by the reviewer, this was previously Figure S7 from the supplementary material). To compare the results from this contrast to conceptual processing across context, we use cosine similarity, and we mention these results in the text. We furthermore show the overlap between the conditions of interest (abstract situated x concrete across context; concrete displaced x abstract across context) in a new figure (Figure 7) to bring out the spatial distribution of overlap more clearly.

We also discussed the extent to which these effects reflect current visual processing only or more general sensory processing in lines 863 – 875 (pg. 33 and 34).

“In considering the impact of visual context on the neural encoding of concepts generally, it is furthermore essential to recognize that the mechanisms observed may extend beyond visual processing to encompass more general sensory processing mechanisms. The human brain is adept at integrating information across sensory modalities to form coherent conceptual representations, a process that is critical for navigating the multimodal nature of real-world experiences (Barsalou, 2008; Smith & Kosslyn, 2007). While our findings highlight the role of visual context in modulating the neural representation of abstract and concrete words, similar effects may be observed in contexts that engage other sensory modalities. For instance, auditory contexts that provide relevant sound cues for certain concepts could potentially influence their neural representation in a manner akin to the visual contexts examined in this study. Future research could explore how different sensory contexts, individually or in combination, contribute to the dynamic neural encoding of concepts, further elucidating the multimodal foundation of semantic processing.”

Overall, the study would benefit from being situated in the literature more, including (a) a more general understanding of the areas involved in semantic processing (including areas proposed to be involved across different sensory modalities and for verbal and nonverbal stimuli), and (b) other differences between abstract and concrete words and whether they can explain the current findings, including other psycholinguistic variables which could be included in the model and the concept of semantic diversity (Hoffman et al.,). It would also be useful to consider whether difficulty effects (or processing effort) could explain some of the regional differences between abstract and concrete words (e.g., the language areas may simply require more of the same processing not more linguistic processing due to their greater reliance on word co-occurrence). Similarly, the findings are not considered in relation to prior comparisons of abstract and concrete words at the level of specific brain regions.

We now present an overview of the areas involved in semantic processing (across different sensory modalities for verbal and nonverbal stimuli) when we first present our results (section: “Conceptual Processing Across Context”).

We looked at surprisal as a potential cofound and found no significant differences between any of the set of words analysed. Mean surprisal of concrete words is 22.19, mean surprisal of abstract words is 21.86. Mean surprisal ratings for concrete situated words are 21.98 bits, 22.02 bits for the displaced concrete words, 22.10 for the situated abstract words and 22.25 for the abstract displaced words. We also calculated the semantic diversity of all sets of words and found now significant differences between the sets. The mean values for each condition are: abstract_high (2.02); abstract_low (1.95); concrete_high (1.88); concrete_low (2.19); abstract_original (1.96); concrete_original (1.92). Hence processing effort related to different predictability (surprisal), or greater semantic diversity cannot explain our findings.

We submit that difficulty effects do not explain any aspects of the activation found for conceptual processing, because we included word frequency in our model as a nuisance regressor and found no significant differences associated with surprisal. Previous work shows that surprisal (Hale, 2001) and word frequency (Brysbaert & New, 2009) are good controls for processing difficulty.

Finally, we added considerations of prior findings comparing abstract and concrete words at the level of specific brain regions to the discussion (section: Conceptual Processing Across Context).

The authors use multiple methods to provide a post hoc interpretation of the areas identified as more involved in concrete, abstract, or both (at different times) words. These are designed to reduce the interpretation bias and improve interpretation, yet they may not successfully do so. These methods do give some evidence that sensory areas are more involved in concrete word processing. However, they are still open to interpretation bias as it is not clear whether all the evidence is consistent with the hypotheses or if this is the best interpretation of individual regions' involvement. This is because the hypotheses are provided at the level of 'sensory' and 'language' areas without further clarification and areas and terms found are simply interpreted as fitting these definitions. For instance, the right IFG is interpreted as a motor area, and therefore sensory as predicted, and the term 'autobiographical memory' is argued to be interoceptive. Language is associated with the 'both' cluster, not the abstract cluster, when abstract >concrete is expected to engage language more. The areas identified for both vs. abstract>concrete are distinguished in the Discussion through the description as semantic vs. language areas, but it is not clear how these are different or defined. Auditory areas appear to be included in the sensory prediction at times and not at others. When they are excluded, the rationale for this is not given. Overall, it is not clear whether all these areas and terms are expected and support the hypotheses. It should be possible to specify specific sensory areas where concrete and abstract words are predicted to be different based on (a) prior comparisons and/or (b) the known locations of sensory areas. Similarly, language or semantic areas could be identified using masks from NeuroSynth or traditional metaanalyses. A language network is presented in Supplementary Figure 7 but not interpreted, and its source is not given.

“The language network” was extracted through neurosynth and projected onto the “overlap” activation map with AFNI. We now specify this in the figure caption.

Alternatively, there could be a greater interpretation of different possible explanations of the regions found with a more comprehensive assessment of the literature. The function of individual regions and the explanation of why many of these areas are interpreted as sensory or language areas are only considered in the Discussion when it could inform whether the hypotheses have been evidenced in the results section.

We added extended considerations of this to the results (as requested by the reviewer) in the section “Conceptual Processing Across Contexts”.

“Consistent with previous studies, we predicted that across naturalistic contexts, concrete and abstract concepts are processed in a separable set of brain regions. To test this, we contrasted concrete and abstract modulators at each time point of the IRF (Figure 3). This showed that concrete produced more modulation than abstract processing in parts of the frontal lobes, including the right posterior inferior frontal gyrus (IFG) and the precentral sulcus (Figure 3, red). Known for its role in language processing and semantic retrieval, the IFG has been hypothesised to be involved in the processing of action-related words and sentences, supporting both semantic decision tasks and the retrieval of lexical semantic information **(**Bookheimer, 2002; Hagoort, 2005). The precentral sulcus is similarly linked to the processing of action verbs and motor-related words (Pulvermüller, 2005). In the temporal lobes, greater modulation occurred in the bilateral transverse temporal gyrus and sulcus, planum polare and temporale. These areas, including primary and secondary auditory cortices, are crucial for phonological and auditory processing, with implications for the processing of sound-related words and environmental sounds (Binder et al., 2000). The superior temporal gyrus (STG) and sulcus (STS) also showed greater modulation for concrete words and these are said to be central to auditory processing and the integration of phonological, syntactic, and semantic information, with a particular role in processing meaningful speech and narratives (Hickok & Poeppel, 2007). In the parietal and occipital lobes, more concrete modulated activity was found bilaterally in the precuneus, which has been associated with visuospatial imagery, episodic memory retrieval, and self-processing operations and has been said to contribute to the visualisation aspects of concrete concepts (Cavanna & Trimble, 2006). More activation was also found in large swaths of the occipital cortices (running into the inferior temporal lobe), and the ventral visual stream. These regions are integral to visual processing, with the ventral stream (including areas like the fusiform gyrus) particularly involved in object recognition and categorization, linking directly to the visual representation of concrete concepts (Martin, 2007). Finally, subcortically, the dorsal and posterior medial cerebellum were more active bilaterally for concrete modulation. Traditionally associated with motor function, some studies also implicate the cerebellum in cognitive and linguistic processing, including the modulation of language and semantic processing through its connections with cerebral cortical areas (Stoodley & Schmahmann, 2009).

Conversely, activation for abstract was greater than concrete words in the following regions (Figure 3, blue): In the frontal lobes, this included right anterior cingulate gyrus, lateral and medial aspects of the superior frontal gyrus. Being involved in cognitive control, decisionmaking, and emotional processing, these areas may contribute to abstract conceptualization by integrating affective and cognitive components (Shenhav et al., 2013). More left frontal activity was found in both lateral and medial prefrontal cortices, and in the orbital gyrus, regions which are key to social cognition, valuation, and decision-making, all domains rich in abstract concepts (Amodio & Frith, 2006). In the parietal lobes, bilateral activity was greater in the angular gyri (AG) and inferior parietal lobules, including the postcentral gyrus. Central to the default mode network, these regions are implicated in a wide range of complex cognitive functions, including semantic processing, abstract thinking, and integrating sensory information with autobiographical memory (Seghier, 2013). In the temporal lobes, activity was restricted to the STS bilaterally, which plays a critical role in the perception of intentionality and social interactions, essential for understanding abstract social concepts (Frith & Frith, 2003). Subcortically, activity was greater, bilaterally, in the anterior thalamus, nucleus accumbens, and left amygdala for abstract modulation. These areas are involved in motivation, reward processing, and the integration of emotional information with memory, relevant for abstract concepts related to emotions and social relations (Haber & Knutson, 2010, Phelps & LeDoux, 2005).

Finally, there was overlap in activity between modulation of both concreteness and abstractness (Figure 3, yellow). The overlap activity is due to the fact that we performed general linear tests for the abstract/concrete contrast at each of the 20 timepoints in our group analysis. Consequently, overlap means that activation in these regions is modulated by both concrete and abstract word processing but at different time-scales. In particular, we find that activity modulation associated with abstractness is generally processed over a longer timeframe (for a comparison of significant timing differences see figure S9). In the frontal, parietal, and temporal lobes, this was primarily in the left IFG, AG, and STG, respectively. Left IFG is prominently involved in semantic processing, particularly in tasks requiring semantic selection and retrieval and has been shown to play a critical role in accessing semantic memory and resolving semantic ambiguities, processes that are inherently timeconsuming and reflective of the extended processing time for abstract concepts (ThompsonSchill et al., 1997; Wagner et al., 2001; Hofman et al., 2015). The STG, particularly its posterior portion, is critical for the comprehension of complex linguistic structures, including narrative and discourse processing. The processing of abstract concepts often necessitates the integration of contextual cues and inferential processing, tasks that engage the STG and may extend the temporal dynamics of semantic processing (Ferstl et al., 2008; Vandenberghe et al., 2002). In the occipital lobe, processing overlapped bilaterally around the calcarine sulcus, which is associated with primary visual processing (Kanwisher et al., 1997; Kosslyn et al., 2001).”

Additionally, these methods attempt to interpret all the clusters found for each contrast in the same way when they may have different roles (e.g., relate to different senses). This is a particular issue for the peaks and valleys method which assesses whether a significantly larger number of clusters is associated with each sensory term for the abstract, concrete, or both conditions than the other conditions. The number of clusters does not seem to be the right measure to compare. Clusters differ in size so the number of clusters does not represent the area within the brain well. Nor is it clear that many brain regions should respond to each sensory term, and not just one per term (whether that is V1 or the entire occipital lobe, for instance). The number of clusters is therefore somewhat arbitrary. This is further complicated by the assessment across 20 time points and the inclusion of the 'both' categories. It would seem more appropriate to see whether each abstract and concrete cluster could be associated with each different sensory term and then summarise these findings rather than assess the number of abstract or concrete clusters found for each independent sensory term. In general, the rationale for the methods used should be provided (including the peak and valley method instead of other possible options e.g., linear regression).

We included an assessment of whether each abstract and concrete cluster could be associated with each different sensory term and then summarised these findings on a participant level in the supplementary material (Figures S3, S4, and S5).

Rationales for the Amplitude Modulated Deconvolution are now provided on page 10 (specifically the first paragraph under “Deconvolution Analysis” in the Methods section) and for the P&V on pages 13, 14 and 15 (under “Peaks and Valley” (particularly the first paragraph) in the Methods section).

The measure of contextual situatedness (how related a spoken word is to the average of the visually presented objects in a scene) is an interesting approach that allows parametric variation within naturalistic stimuli, which is a potential strength of the study. This measure appears to vary little between objects that are present (e.g., animal or room), and those that are strongly (e.g., monitor) or weakly related (e.g., science). Additional information validating this measure may be useful, as would consideration of the range of values and whether the split between situated (c > 0.6) and displaced words (c < 0.4) is sufficient.

The main validation of our measure of contextual situatedness derives from the high accuracy and reliability of CNNs in object detection and recognition tasks, as demonstrated in numerous benchmarks and real-world applications.

One reason for low variability in our measure of contextual situatedness is the fact that we compared the GloVe vector of each word of interest with an average GloVe vector of all object-words referring to objects present in 56 frames (~300 objects on average). This means that a lot of variability in similarity measures between individual object-words and the word of interest is averaged out. Notwithstanding the resulting low variability of our measure, we thought that this would be the more conservative approach, as even small differences between individual measures (e.g. 0.4 vs 0.6) would constitute a strong difference on average (across the 300 objects per context window). Therefore, this split ensures a sufficient distinction between words that are strongly related to their visual context and those that are not – which in turn allows us to properly investigate the impact of contextual relevance on conceptual processing.

Finally, the study assessed the relation of spoken concrete or abstract words to brain activity at different time points. The visual scene was always assessed using the 2 seconds before the word, while the neural effects of the word were assessed every second after the presentation for 20 seconds. This could be a strength of the study, however almost no temporal information was provided. The clusters shown have different timings, but this information is not presented in any way. Giving more temporal information in the results could help to both validate this approach and show when these areas are involved in abstract or concrete word processing.

We provide more information on the temporal differences of when clusters are involved in processing concrete and abstract concepts in the supplementary material (Figure S9) and refer to this information where relevant in the Methods and Results sections.

Additionally, no rationale was given for this long timeframe which is far greater than the time needed to process the word, and long after the presence of the visual context assessed (and therefore ignores the present visual context).

The 20-second timeframe for our deconvolution analysis is justified by several considerations. Firstly, the hemodynamic response function (HRF) is known to vary both across individuals and within different regions of the brain. To accommodate this variability and capture the full breadth of the HRF, including its rise, peak, and return to baseline, a longer timeframe is often necessary. The 20-second window ensures that we do not prematurely truncate the HRF, which could lead to inaccurate estimations of neural activity related to the processing of words. Secondly and related to this point, unlike model-based approaches that assume a canonical HRF shape, our deconvolution analysis does not impose a predefined form on the HRF, instead reconstructing the HRF from the data itself – for this, a longer time-frame is advantageous to get a better estimation of the true HRF. Finally, and related to this point, the use of the 'Csplin' function in our analysis provides a flexible set of basis functions for deconvolution, allowing for a more fine-grained and precise estimation of the HRF across this extended timeframe. The 'Csplin' function offers more interpolation between time points, which is particularly advantageous for capturing the nuances of the HRF as it unfolds over a longer time-frame.

Although we use a 20-second timeframe for the deconvolution analysis to capture the full HRF, the analysis is still time-locked to the onset of each visual stimulus. This ensures that the initial stages of the HRF are directly tied to the moment the word is presented, thus incorporating the immediate visual context. We furthermore include variables that represent aspects of the visual context at the time of word presentation in our models (e.g luminance) and control for motion (optical flow), colour saturation and spatial frequency of immediate visual context.

**Reviewer #3 (Public Review):**
The context measure is interesting, but I'm not convinced that it's capturing what the authors intended. In analysing the neural response to a single word, the authors are presuming that they have isolated the window in which that concept is processed and the observed activation corresponds to the neural representation of that word given the prior context. I question to what extent this assumption holds true in a narrative when co-articulation blurs the boundaries between words and when rapid context integration is occurring.

We appreciate the reviewer's critical perspective on the contextual measure employed in our study. We agree that the dynamic and continuous nature of narrative comprehension poses challenges for isolating the neural response to individual words. However, the use of an amplitude modulated deconvolution analysis, particularly with the CSPLIN function, is a methodological choice to specifically address these challenges. Deconvolution allows us to estimate the hemodynamic response function (HRF) without assuming its canonical shape, capturing nuances in the BOLD signal that may reflect the integration of rapid contextual shifts only beyond the average modulation of the BOLD signal. The CSPLIN function further refines this approach by offering a flexible basis set for modelling the HRF and by providing a detailed temporal resolution that can adapt to the variance in individual responses.

Our choice of a 20-second window is informed by the need to encompass not just the immediate response to a word but also the extended integration of the contextual information. This is consistent with evidence indicating that the brain integrates information over longer timescales when processing language in context (Hasson et al., 2015). The neural representation of a word is not a static snapshot but a dynamic process that evolves with the unfolding narrative.

Further, the authors define context based on the preceding visual information. I'm not sure that this is a strong manipulation of the narrative context, although I agree that it captures some of the local context. It is maybe not surprising that if a word, abstract or concrete, has a strong association with the preceding visual information then activation in the occipital cortex is observed. I also wonder if the effects being captured have less to do with concrete and abstract concepts and more to do with the specific context the displaced condition captures during a multimodal viewing paradigm. If the visual information is less related to the verbal content, the viewer might process those narrative moments differently regardless of whether the subsequent word is concrete or abstract. I think the claims could be tailored to focus less generally on context and more specifically on how visually presented objects, which contribute to the ongoing context of a multimodal narrative, influence the subsequent processing of abstract and concrete concepts.

The context measure, though admittedly a simplification, is designed to capture the local visual context preceding word presentation. By using high-confidence visual recognition models, we ensure that the visual information is reliably extracted and reflects objects that have a strong likelihood of influencing the processing of subsequent words. We acknowledge that this does not capture the full richness of narrative context; however, it provides a quantifiable and consistent measure of the immediate visual environment, which is an important aspect of context in naturalistic language comprehension.

With regards to the effects observed in the occipital cortex, we posit that while some activation might be attributable to the visual features of the narrative, our findings also reflect the influence of these features on conceptual processing. This is especially because our analysis only looks at the modulation of the HRF amplitude beyond the average response (so also beyond the average visual response) when contrasting between conditions of high and low visual-contextual association with important (audio-visual) control variables included in the model.

Lastly, we concur that both concrete and abstract words are processed within a multimodal narrative, which could influence their neural representation. We believe our approach captures a meaningful aspect of this processing, and we have refined our claims to specify the influence of visually presented objects on the processing of abstract and concrete concepts, rather than making broader assertions about multimodal context. We also highlight several other signals (e.g. auditory) that could influence processing.

**Recommendations for the authors:**

**Reviewer #1 (Recommendations For The Authors):**
(1) The approach taken here requires a lot of manual variable selection and seems a bit roundabout. Why not build an encoding model that can predict the BOLD time course of each voxel in a participant from the feature-of-interest like valence etc. and then analyze if (1) certain features better predict activity in a specific region (2) the predicted responses/regression parameters are more positive (peaks) or more negative (valleys) for certain features in a specific brain region (3) maybe even use contextual features use a large language model and then per word (like "truth") analyze where the predicted responses diverge based on the associated context. This seems like a simpler approach than having multiple stages of analysis.

It is not clear to us why an encoding model would be more suitable for answering the question at hand (especially given that we tried to clarify concerns about non-linear relationships between variables). On the contrary, fitting a regression model to each individual voxel has several drawbacks. First, encoding models are prone to over-estimate effect sizes (Naselaris et al., 2011). Second, encoding models are not good at explaining group-level effects due to high variability between individual participants (Turner et al., 2018). We would also like to point out that an encoding model using features of a text-based LLM would not address the visual context question - unless the LLM was multimodal. Multimodal LLMs are a very recent research development in Artificial Intelligence, however, and models like LLaMA (adapter), Google’s Gemini, etc. are not truly multimodal in the sense that would be useful for this study, because they are first trained on text and later injected with visual data. This relates to our concern that the reviewer may have misunderstood that we are interested in purely visual context of words (not linguistic context).

(2) In multiple analyses, a subset of the selected words is sampled to create a balanced set between the abstract and concrete categories. Do the authors show standard deviation across these sets?

For the subset of words used in the context-based analyses, we give mean ratings of concreteness, log frequency and length and conduct a t-test to show that these variables are not significantly different between the sets. We also included the psycholinguistic control variables surprisal and semantic diversity, as well as the visual variables motion (optical flow), colour saturation and spatial frequency.

**Reviewer #2 (Recommendations For The Authors):**
Figures S3-5 are central to the argument and should be in the main text (potentially combined).

These have been added to the main text

S5 says the top 3 terms are DMN (and not semantic control), but the text suggests the r value is higher for 'semantic control' than 'DMN'?

Fixed this in the text, the caption now reads:

“This was confirmed by using the neurosynth decoder on the unthresholded brain image - top keywords were “Semantic Control” and “DMN”.”

Fig. S7 is very hard to see due to the use of grey on grey. Not used for great effect in the final sentence, but should be used to help interpret areas in the results section (if useful). It has not been specified how the 'language network' has been identified/defined here.

We altered the contrast in the figure to make boundaries more visible and specified how the language network was identified in the figure caption.

In the Results 'This showed that concrete produced more modulation than abstract modulation in the frontal lobes,' should be parts of /some of the frontal lobes as this isn't true overall.

Fixed this in the text.

There are some grammatical errors and lack of clarity in the context comparison section of the results.

Fixed these in the text.

**Reviewer #3 (Recommendations For The Authors):**
• The analysis code should be shared on the github page prior to peer review.

The code is now shared under: https://github.com/ViktorKewenig/Naturalistic_Encoding_Concepts

• At several points throughout the methods section, information was referred to that had not yet been described. Reordering the presentation of this information would greatly improve interpretability. A couple of examples of this are provided below.Deconvolution Analysis: the use of amplitude modulation regression was introduced prior to a discussion of using the TENT function to estimate the shape of the HRF. This was then followed by a discussion of the general benefits of amplitude modulation. Only after these paragraphs are the modulators/model structure described. Moving this information to the beginning of the section would make the analysis clearer from the onset.

Fixed this in the text

Peak and Valley Analysis: the hypotheses regarding the sensory-motor features and experiential features are provided prior to describing how these features were extracted from the data (e.g., using the Lancaster norms).

Fixed this in the text.

• The justification for and description of the IRF approach seems overdone considering the timing differences are not analyzed further or discussed.

We now present a further discussion of timing differences in the supplementary material.

• The need and suitability of the cluster simulation method as implemented were not clear. The resulting maps were thresholded at 9 different p values and then combined, and an arbitrary cluster threshold of 20 voxels was then applied. Why not use the standard approach of selecting the significance threshold and corresponding cluster size threshold from the ClustSim table?

We extracted the original clusters at 9 different p values with the corresponding cluster size from the ClustSim table, then only included clusters that were bigger than 20 voxels.

• Why was the center of mass used instead of the peak voxel?

Peak voxel analysis can be sensitive to noise and may not reliably represent the region's activation pattern, especially in naturalistic imaging data where signal fluctuations are more variable and outliers more frequent. The centre of mass provides a more stable and representative measure of the underlying neural activity. Another reason for using the center of mass is that it better represents the anatomical distribution of the data, especially in large clusters with more than 100 voxels where peak voxels are often located at the periphery.

• Figure 1 seems to reference a different Figure 1 that shows the abstract, concrete, and overlap clusters of activity (currently Figure 3).

Fixed this in the text.

• Table S1 seems to have the "Touch" dimension repeated twice with different statistics reported.

Fixed this in the text, the second mention of the dimension “touch” was wrong.

• It appears from the supplemental files that the Peaks and Valley analysis produces different results at different lag times. This might be expected but it's not clear why the results presented in the main text were chosen over those in the supplemental materials.

The results in the main text were chosen over those in the supplementary material, because the HRF is said to peak at 5s after stimulus onset. We added a specification of this rational to the “2. Peak and Valley Analysis” subsection in the Methods section.

References (in order of appearance)

(1) Neumann J, Lohmann G, Zysset S, von Cramon DY. Within-subject variability of BOLD response dynamics. Neuroimage. 2003 Jul;19(3):784-96. doi: 10.1016/s10538119(03)00177-0. PMID: 12880807.

(2) Handwerker DA, Ollinger JM, D'Esposito M. Variation of BOLD hemodynamic responses across subjects and brain regions and their effects on statistical analyses. Neuroimage. 2004 Apr;21(4):1639-51. doi: 10.1016/j.neuroimage.2003.11.029. PMID: 15050587.

(3) Binder JR, Westbury CF, McKiernan KA, Possing ET, Medler DA. Distinct brain systems for processing concrete and abstract concepts. J Cogn Neurosci. 2005 Jun;17(6):90517. doi: 10.1162/0898929054021102. PMID: 16021798

(4) Bucur, M., Papagno, C. An ALE meta-analytical review of the neural correlates of abstract and concrete words. Sci Rep 11, 15727 (2021). https://doi.org/10.1038/s41598-021-94506-9.

(5) Hale., J. 2001. A probabilistic earley parser as a psycholinguistic model. In Proceedings of the second meeting of the North American Chapter of the Association for Computational Linguistics on Language technologies (NAACL '01). Association for Computational Linguistics, USA, 1–8. https://doi.org/10.3115/1073336.1073357.

(6) Brysbaert, M., New, B. Moving beyond Kučera and Francis: A critical evaluation of current word frequency norms and the introduction of a new and improved word frequency measure for American English. Behavior Research Methods 41, 977–990 (2009). https://doi.org/10.3758/BRM.41.4.977.

(7) Hasson, U., Nir, Y., Levy, I., Fuhrmann, G., & Malach, R. (2004). Intersubject Synchronization of Cortical Activity During Natural Vision. *Science*, *303*(5664), 6.

(8) Naselaris T, Kay KN, Nishimoto S, Gallant JL. Encoding and decoding in fMRI. Neuroimage. 2011 May 15;56(2):400-10. doi: 10.1016/j.neuroimage.2010.07.073. Epub 2010 Aug 4. PMID: 20691790; PMCID: PMC3037423.

(9) Turner BO, Paul EJ, Miller MB, Barbey AK. Small sample sizes reduce the replicability of task-based fMRI studies. Commun Biol. 2018 Jun 7;1:62. doi: 10.1038/s42003-0180073-z. PMID: 30271944; PMCID: PMC6123695.

(10) He, K., Zhang, Y., Ren, S., & Sun, J. (2015). Deep Residual Learning for Image Recognition. *Bioarchive (Tech Report)*. https://doi.org/10.48550/arXiv.1512.03385.

(11) Hasson, U., & Egidi, G. (2015). What are naturalistic comprehension paradigms teaching us about language? In R. M. Willems (Ed.), *Cognitive neuroscience of natural language use* (pp. 228–255). Cambridge University Press. https://doi.org/10.1017/CBO9781107323667.011.